# Nuclear Hsp104 safeguards the dormant translation machinery during quiescence

Verena Kohler [1,2,3], Andreas Kohler [2,4,5], Lisa Larsson Berglund[6], Xinxin Hao [7], Sarah Gersing[8], Axel Imhof [9], Thomas Nyström [7], Johanna L. Höög [6], Martin Ott[4,10], Claes Andréasson [1] ✉ & Sabrina Büttner [1] ✉

The resilience of cellular proteostasis declines with age, which drives protein aggregation and compromises viability. The nucleus has emerged as a key quality control compartment that handles misfolded proteins produced by the cytosolic protein biosynthesis system. Here, we find that age-associated metabolic cues target the yeast protein disaggregase Hsp104 to the nucleus to maintain a functional nuclear proteome during quiescence. The switch to respiratory metabolism and the accompanying decrease in translation rates direct cytosolic Hsp104 to the nucleus to interact with latent translation initiation factor eIF2 and to suppress protein aggregation. Hindering Hsp104 from entering the nucleus in quiescent cells results in delayed re-entry into the cell cycle due to compromised resumption of protein synthesis. In sum, we report that cytosolic-nuclear partitioning of the Hsp104 disaggregase is a critical mechanism to protect the latent protein synthesis machinery during quiescence in yeast, ensuring the rapid restart of translation once nutrients are replenished.

Cellular proteostasis is maintained by a complex network of factors that coordinate protein biosynthesis, structural maintenance and proteolytic turnover[1–5]. This network safeguards the functionality of the proteome and is crucial for cellular fitness and longevity[3,6–8]. With age, the functionality of the proteostasis network progressively declines, resulting in the build-up of misfolded proteins. In humans, compromised proteostasis is linked to various pathologies, including cancer and neurodegenerative disorders, and the accumulation of aggregated proteins represents a universal hallmark of aging[3,9–11]. Protein quality control (PQC) machineries are critical components of the proteostasis network that selectively recognize misfolded proteins

and either refold or degrade these aberrant species, hence suppressing the build-up of potentially toxic and aggregation-prone proteins[7,8,12].

Proteostasis is challenged by the constant production of unfolded proteins by the translating ribosome as well as by misfolded proteins derived from faulty gene expression, age-induced damage and environmental stress[3,9,13,14]. Translation rates are intimately linked to cell growth and metabolic cues. In yeast cells growing exponentially under nutrient-rich conditions, the proteostasis network is mainly challenged by the high rates of protein synthesis[4]. Upon nutrient depletion and the diauxic shift, which entails the switch from fermentative to respiratory metabolism, translation rates decrease, cells exit the cell

[1]Department of Molecular Biosciences, The Wenner-Gren Institute, Stockholm University, 10691 Stockholm, Sweden. [2]Institute of Molecular Biosciences, University of Graz, 8010 Graz, Austria. [3]Department of Molecular Biology, Umeå University, 90187 Umeå, Sweden. [4]Department of Biochemistry and Biophysics, Stockholm University, 10691 Stockholm, Sweden. [5]Department of Medical Biochemistry and Biophysics, Umeå University, 90187 Umeå, Sweden. [6]Department of Chemistry and Molecular Biology, University of Gothenburg, 40530 Gothenburg, Sweden. [7]Department of Microbiology and Immunology, University of Gothenburg, 40530 Gothenburg, Sweden. [8]The Linderstrøm-Lang Centre for Protein Science, Department of Biology, University of Copenhagen, 1165 Copenhagen, Denmark. [9]Biomedical Center Munich, Faculty of Medicine, Ludwig Maximilian University of Munich, 82152 Planegg-Martinsried, Germany. [10]Department of Medical Biochemistry and Cell Biology, University of Gothenburg, 40530 Gothenburg, Sweden. ✉e-mail: claes.andreasson@su.se; sabrina.buettner@su.se

cycle and ultimately enter into a non-dividing state termed quiescence, which is characterized by low metabolic activity[15–17]. This is accompanied by the progressive reallocation of resources to proteome maintenance and stress damage control. Cellular remodeling includes major rearrangements of the proteostasis network, such as the formation of membrane-less cytoplasmic proteasome storage granules, the sequestration of fundamental parts of the cytosolic translation machinery into stress granules and P-bodies as well as the degradation of ribosomes[15,16,18–27]. Nutrient replenishment rapidly reactivates the translation machinery and cells exit quiescence to re-enter the cell cycle[16,17]. At this point, the fundamental translation machinery as well as the proteostasis network need to be primed for this challenging pulse of immediate protein biosynthesis. Thus, cells need to ensure that misfolded and aggregated proteins that tie up proteostasis factors do not accumulate during aging.

Aggregated proteins are disentangled by dedicated ATP-consuming machines called protein disaggregases, enabling their subsequent rescue by refolding or removal by degradation via the ubiquitin-proteasome system[28–32]. In yeast, the disaggregation machinery is centered around the AAA+ ATPase Hsp104, a hexameric disaggregase that threads trapped polypeptides through its central pore, resolving a wide range of aberrant protein species. Hsp104 is recruited to the aggregate surface by Hsp70 chaperones that also activate its ATP-dependent disaggregase activity[33–35]. Hsp104 expression is induced by the stress response transcription factors Msn2/4[36–39] that reprogram gene expression when cells undergo the diauxic shift, suggesting an important function for protein disaggregation during aging and quiescence. Nevertheless, functional studies of the Hsp104 disaggregase as well as the general proteostasis network have almost exclusively been focused on transiently stressed and dividing cells under nutrient-rich conditions[5,33,34,37,39–43].

The nucleus has emerged as a hub for PQC and the handling of misfolded and aggregated proteins. Misfolded proteins can be targeted to the nucleus and sequestered into an aggregate-like assembly called the intranuclear quality control compartment (INQ)[44]. The nuclear targeting of misfolded proteins as well as the deposition of these species into INQ depends on key proteostasis factors[44–48]. Yet, it remains poorly understood how the proper subcellular localization of these factors is orchestrated by age-dependent metabolic cues to safeguard rapid cytosolic translation in dividing cells and nuclear proteostasis upon cell cycle exit.

In this study, we utilize high-content microscopy screening and quantitative proteomics to establish a function of nuclear Hsp104 during aging. We find that the cytosolic-nuclear partitioning of Hsp104 is critical to maintain nuclear proteostasis in quiescent cells. Whereas high global translation rates in rapidly growing cells constrain Hsp104 to the cytosol, a decrease in protein biosynthesis re-directs the disaggregase to the nucleus dependent on a specific C-terminal motif. Nuclear Hsp104 interacts with eIF2 translation initiation complexes and suppresses protein aggregation. This protects the dormant translation machinery from age-induced damage, enabling the rapid resumption of protein synthesis upon re-entry into the cell cycle.

## Results

### The disaggregase Hsp104 is targeted to the nucleus when cells age

To explore how the proteostasis network adapts to changing cellular needs with age, we assessed the subcellular localization of key PQC factors in yeast, including Hsp70 (yeast Ssa1, Ssa4 and Ssb1), Hsp90 (yeast Hsp82), and the major disaggregase Hsp104 using GFP chimeras. In young, dividing cells, all PQC factors were evenly distributed throughout the cell (Fig. 1a, Supplementary Fig. 1). In aged, non-dividing cells, Hsp104 and Hsp82 were re-directed to the nucleus, while Hsp70 remained evenly distributed and decorated cytosolic aggregates forming with age (Fig. 1a, Supplementary Fig. 1). Yeast cells

generate ATP via fermentation in glucose-rich medium but switch to respiration when glucose is exhausted. Analyzing cells under strictly respiratory conditions (glycerol) revealed that metabolic cues differentially regulate the nuclear accumulation of Hsp104 and Hsp82, as only Hsp104 was targeted to the nucleus in respiring cells (Fig. 1b). Visualizing the nuclear membrane showed that Hsp104 localized evenly throughout the nucleus and not to a distinct sub-nuclear compartment (Fig. 1c). After entry into stationary phase (48 h), around 80% of cells displayed Hsp104 in the nucleus (Fig. 1d, e). Progressive nuclear accumulation of Hsp104 over time was detectable when comparing the signal intensity ratio between the nucleus and cytoplasm (Fig. 1f). After extended aging (7 days), Hsp104 decorated specific nuclear sub-compartments, suggesting it targets aggregates forming in the nucleus with cellular age (Fig. 1d). Transmission electron microscopy (TEM) combined with immunogold labeling against native Hsp104 confirmed the age-induced nuclear accumulation (Fig. 1g). As expected from the increased Hsp104 expression levels in stationary phase[39], we found an overall increased Hsp104 labeling density in aged cells (Fig. 1g). Thus, Hsp104 is mainly cytosolic in young, growing cells and is targeted to the nucleus when cells age or adjust to respiratory metabolism.

### Nuclear targeting of Hsp104 depends on a motif in its flexible C-terminal tail

Comparing the different genetic backgrounds of S288C and W303 strains confirmed the nuclear accumulation of Hsp104 that we originally observed in BY4741 (Fig. 2a, b). In contrast, the Hsp104[GFP] variant from the commercially available genome-wide Yeast GFP clone collection[49] failed to target the nucleus in aged cells despite its BY4741 genetic background (Fig. 2a, b; Supplementary Fig. 2a). The absence of nuclear targeting was found to be dominant upon mating the commercial Yeast GFP clone with the parental strain background (Supplementary Fig. 2b). Sequencing revealed a single methionine to valine substitution (M901V) localized to the C-terminal tail of Hsp104, a suggested binding site for tetratricopeptide repeat (TPR) proteins[35,50], in the strain from the Yeast GFP clone collection (Fig. 2c). Importantly, introducing the M901V substitution (Hsp104*) or deleting the last 8 amino acids from the C-terminal tail (Hsp104[ΔC]) prevented the age-induced nuclear targeting of Hsp104 (Fig. 2d, e). Immunoblot analysis revealed comparable protein levels of Hsp104 and Hsp104* over time, indicating that the nuclear accumulation was indeed due to relocalization and not degradation of Hsp104 specifically in the cytosol (Fig. 2f). Quantitative analysis of the nuclear accumulation showed that Hsp104* was even excluded from the nucleus (Fig. 2g, h). Notably, fusing the last 8 amino acids of Hsp104, corresponding to the TPR protein binding motif, to GFP was sufficient to target the chimera to the nucleus in an age-dependent manner (Fig. 2i, j). The M901V mutation abrogated nuclear accumulation. Hence, the short C-terminal motif is required and sufficient for the age-induced targeting of Hsp104 to the nucleus.

### Mitochondrial respiration is critical for nuclear accumulation of Hsp104

To gain mechanistic insights into Hsp104 nuclear targeting, we constructed a genome-wide deletion library endogenously equipped with Hsp104[GFP] and screened for age-dependent nuclear localization using high-content microscopy. We scored the gene deletion mutants based on the nucleus-cytoplasm ratio of Hsp104[GFP] after 2 days of growth (Fig. 3a). Functional enrichment analysis of mutants that displayed decreased nuclear Hsp104 accumulation using STRING analysis and Markov algorithm clustering identified genes associated with mitochondrial function, translation and nuclear transport (Fig. 3b; Supplementary Fig. 3a). The identified nuclear transport machinery included Nup133, Pom33, Pom34, Rtn1, Pom152, Apq12 and Kap120, which are well-defined nuclear import factors and components of the

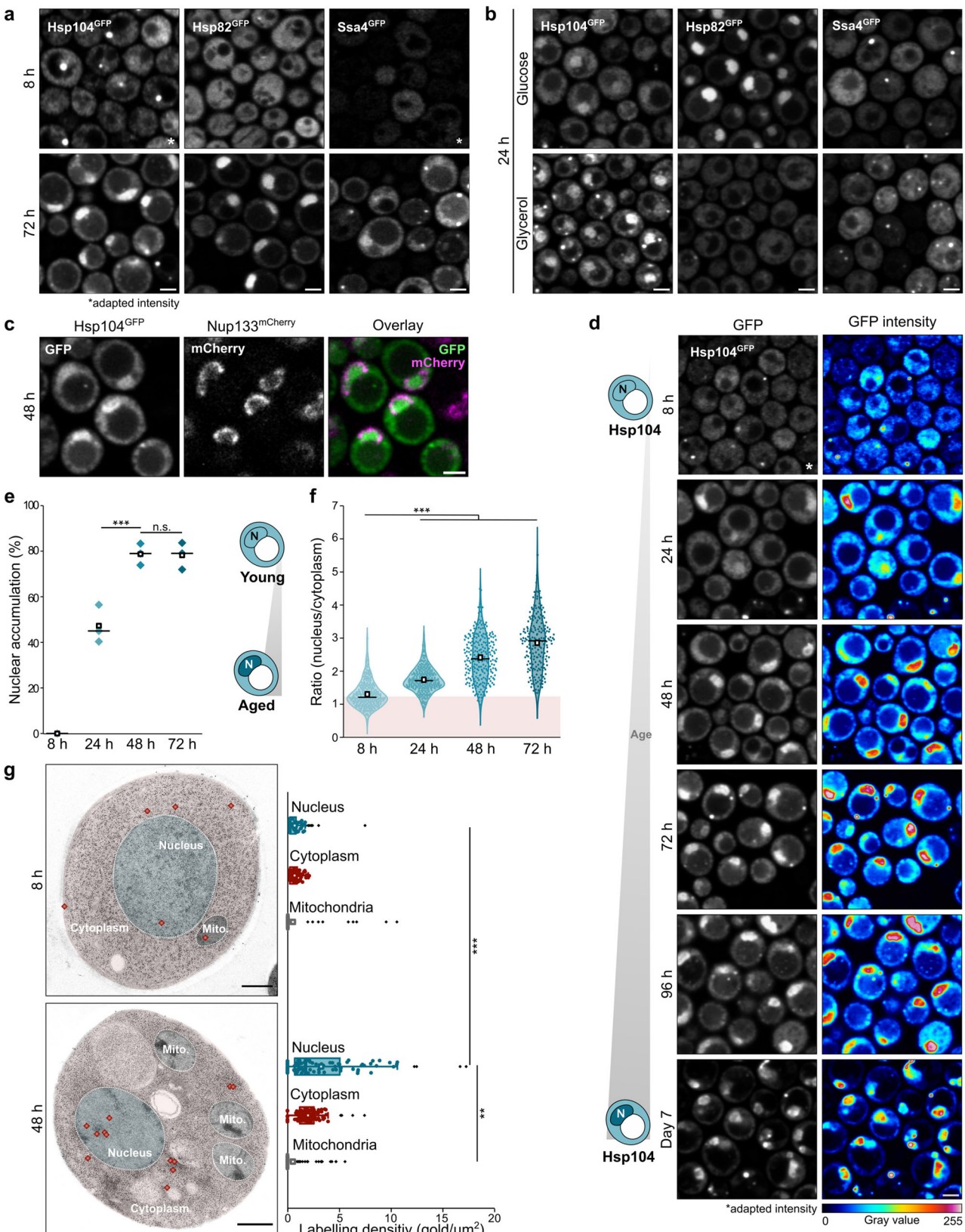

nuclear pore complex[51,52]. We reasoned that mitochondrial respiration and cytosolic translation regulate the nuclear targeting machinery of Hsp104. Indeed, quantitative analysis revealed a strong decrease of cells with nuclear Hsp104 accumulation upon genetic inactivation of respiration via deletion of *COX12* or *MIP1*[53,54] (Fig. 3c, d). Likewise, restricting oxygen availability through non-aerated liquid cultivation,

which reduces respiratory activity and mitochondrial biogenesis, prevented age-induced nuclear accumulation of Hsp104 (Fig. 3e). Vice versa, under respiratory growth conditions, Hsp104 localized to the nucleus already in young, growing cells (Fig. 3f, g). Notably, acute pharmacological inhibition of respiration by antimycin A in cells approaching the stationary phase triggered relocalization of nuclear

**Fig. 1 | The disaggregase Hsp104 is targeted to the nucleus when cells age.**
**a** Micrographs of cells harboring endogenously GFP-tagged Hsp104, Hsp90 (yeast Hsp82) and Hsp70 (yeast Ssa4), respectively, visualized at 8 h and 72 h. The signal intensity of GFP-tagged Hsp104 and Ssa4 at 8 h was increased to ensure visibility (*). Scale bar: 2 μm. **b** Micrographs of cells described in (**a**) grown for 24 h on glucose or glycerol. Scale bar: 2 μm. **c** Micrographs of cells endogenously expressing Hsp104$^{GFP}$ and Nup133$^{mCherry}$ to visualize the nucleus after culturing for 48 h on glucose. Scale bar: 2 μm. **d** Micrographs of cells endogenously expressing Hsp104$^{GFP}$, visualized at indicated time points. Schematic drawings illustrate major phenotypes. *N* = Nucleus. The signal intensity of Hsp104$^{GFP}$ at 8 h was increased to ensure visibility. Scale bar: 2 μm. **e** Quantification of nuclear accumulation of Hsp104$^{GFP}$ in cells described in (**d**). Dot plots with mean (square) and median (line). Each dot represents one biological replicate (8 h: 444 cells; 24 h: 347 cells; 48 h: 380 cells; 72 h: 341 cells). Schematic drawings illustrate major phenotypes. *N* = Nucleus.

**f** Quantification of the mean GFP intensity ratio between nucleus and cytoplasm of cells described in (**d**). Violin plots with mean (square) and median (line). Each dot represents one cell (8 h: 267 cells; 24 h: 275 cells; 48 h: 252 cells; 72 h: 203 cells). Measurements were taken from micrographs of 3 biological replicates. **g** Hsp104 immunogold-labeling of untagged cells at indicated time points. Representative transmission electron micrographs and quantification of labeling density are depicted. The nucleus (turquoise), cytoplasm (red) and mitochondria (Mito., gray) are highlighted and immunogold signal is depicted as red diamonds. Mitochondria served as negative control. Box plots with mean (square) and median (line) and whiskers with minima and maxima within 1.5 interquartile range are shown. 100 cells were analyzed per time point. Scale bar: 500 μm. n.s.: not significant ($p \geq 0.05$); **$p < 0.01$; ***$p < 0.001$. Source data are provided as a Source Data file. See Supplementary Table 3 for details on statistical analyses.

---

Hsp104 into the cytosol, which was accompanied by the formation of cytosolic aggregates (Fig. 3h). The key stress transcription factors activated during respiratory growth and entry into stationary phase, Msn2/4[36–39], were not required to target Hsp104 to the nucleus (Supplementary Fig. 3b). Loss of nuclear Hsp104 did not affect Msn2/4 activity (Supplementary Fig. 3c). Collectively, the genome-wide screen revealed the presence of a Hsp104 nuclear targeting machinery and identified mitochondrial respiration, but not its associated stress response signaling, as a key metabolic cue that controls nuclear relocalization.

## Nuclear Hsp104 interacts with translation-associated factors in aged cells

We capitalized on the Hsp104* variant that is excluded from the nucleus to determine the interactome of cytosolic and nuclear Hsp104 using two different approaches. First, we applied biotin proximity-labeling (BioID), where a promiscuous biotin ligase variant (BirA*) was fused to either Hsp104 or Hsp104*, and proximal proteins were identified by mass spectrometry (Fig. 4a). In a parallel approach, we performed co-immunoprecipitation experiments based on GFP nanobodies (Fig. 4b). Cells were aged for 48 h to allow a clear distinction between the interaction networks of cytosolic Hsp104* and nuclear Hsp104. For both approaches, proteins significantly enriched in the nuclear Hsp104 (wild type) interactome were subjected to STRING analysis, revealing translation-associated factors as the most prominently enriched cluster (Fig. 4c, d). This included structural ribosomal proteins as well as factors regulating translation (Fig. 4c, d, Supplementary Fig. 4a–d). Notably, both proteomic approaches identified major translation initiation factors (eIF2α/Sui2, eIF2γ/Gcd11, eIF4A/Tif1 and eIF4E/Cdc33), translation elongation factors (eEF1A/Tef1, eEF1B/Tef4 and eEF2/Eft1) as well as several aminoacyl-tRNA-synthetases to specifically associate with nuclear Hsp104 (Fig. 4d). Thus, nuclear Hsp104 in aged cells specifically interacts with ribosomal proteins and, more broadly, with translation-associated factors.

## The translation initiation factor eIF2 interacts with nuclear Hsp104

To identify the interactors that are specific for the C-terminal recognition motif near methionine 901 of Hsp104, we introduced the UV-crosslinking amino acid derivative p-benzoyl-L-phenylalanine (Bpa)[55] at amino acid position 897 by stop codon suppression. After UV exposure of cells expressing the wild type Hsp104 or Hsp104*, containing the M901V mutation, we performed affinity purification and mass spectrometric analysis (Fig. 5a). The factors crosslinking specifically with the wild type C-terminal tail included two STRING enrichment clusters: translation-associated factors and proteins involved in proteasomal degradation (Fig. 5b and Supplementary Fig. 5a). We again identified structural ribosomal proteins as well as subunits of the heterotrimeric translation initiation factor eIF2, including eIF2α/Sui2 and eIF2γ/Gcd11 and its associated GTPase

activating protein eIF5/Tif5. Though eIF2β/Sui3 was not present in our analyses, we detected an interaction when performing pulldown experiments coupled with immunoblot analyses (Supplementary Fig. 5c). Revisiting the proteomic profiling of the Hsp104 interactome obtained via BioID and CoIP, we found eIF2α/Sui2 and eIF2γ/Gcd11 to be present in all three proteomic analyses, while eIF5/Tif5 was present in two out of three (Fig. 5c, Supplementary Fig. 5b). Since specifically nuclear Hsp104 interacts with these eIF2 subunits, we speculated on their presence in the nucleus of aged cells. To directly test this hypothesis, we assessed the localization of Sui2$^{mCherry}$ in young versus old cells. This revealed that eIF2α/Sui2 was evenly distributed in young, dividing cells but progressively accumulated in the nucleus as cells aged, resembling the subcellular distribution of Hsp104 (Fig. 5d, e and Supplementary Fig 5d). Interestingly, we observed significantly reduced nuclear targeting of Sui2 in aged Hsp104* cells, assessed via determination of the nuclear/cytoplasmic intensity ratios as well as by quantification of the fraction of cells with nuclear accumulation of Sui2 (Fig. 5d-g and Supplementary Fig. 5d). Sui2 protein levels were comparable in cells expressing either Hsp104 or Hsp104* during aging, ruling out that the changed localization is a consequence of altered expression levels (Fig. 5h). In sum, nuclear Hsp104 interacts with eIF2α/Sui2 and regulates its nucleo-cytoplasmic partitioning in aging cells.

## Reduced translation induces nuclear accumulation of Hsp104 independent of its disaggregation activity

We investigated how protein synthesis in general and eIF2 in particular impacted on the nucleo-cytoplasmic partitioning of Hsp104. First, we depleted eIF2α/Sui2 or its GTPase-activating protein eIF5/Tif5 using the glucose-repressible *GAL*-promoter. This depletion did not impair nuclear targeting of Hsp104, ruling out the possibility that these factors are part of a nuclear targeting mechanism. Instead, the depletion prominently triggered the targeting of Hsp104 to the nucleus even in young cells (Fig. 6a, b). This raised the possibility that the inhibition of translation serves as a cue for routing Hsp104 to the nucleus. Indeed, translation rates and levels of ribosomes are drastically reduced when cells enter the stationary phase[56,57], which coincides with the observed nuclear Hsp104 accumulation. Refeeding starved cells with glucose, which induces rapid translation, resulted in the exit of a significant fraction of Hsp104 from the nucleus within 1 h (Fig. 6c, d). To test whether this depends on the restart of translation, we aged cells with *GAL*-promoter regulated expression of Sui2 and Tif5 in media containing galactose, allowing for initial expression of Sui2 and Tif5 during exponential growth until galactose was depleted. In aged cells depleted of Sui2 or Tif5, the addition of glucose did not trigger the nuclear exit of Hsp104 (Fig. 6c, d), suggesting that translational restart serves as a signal for Hsp104 mobilization from the nucleus. Next, we used cycloheximide to acutely inhibit bulk translation in young, dividing cells. Hsp104 was rapidly targeted to the nucleus within the first 45 min after treatment (Fig. 6e, f). As cycloheximide inhibits the elongation

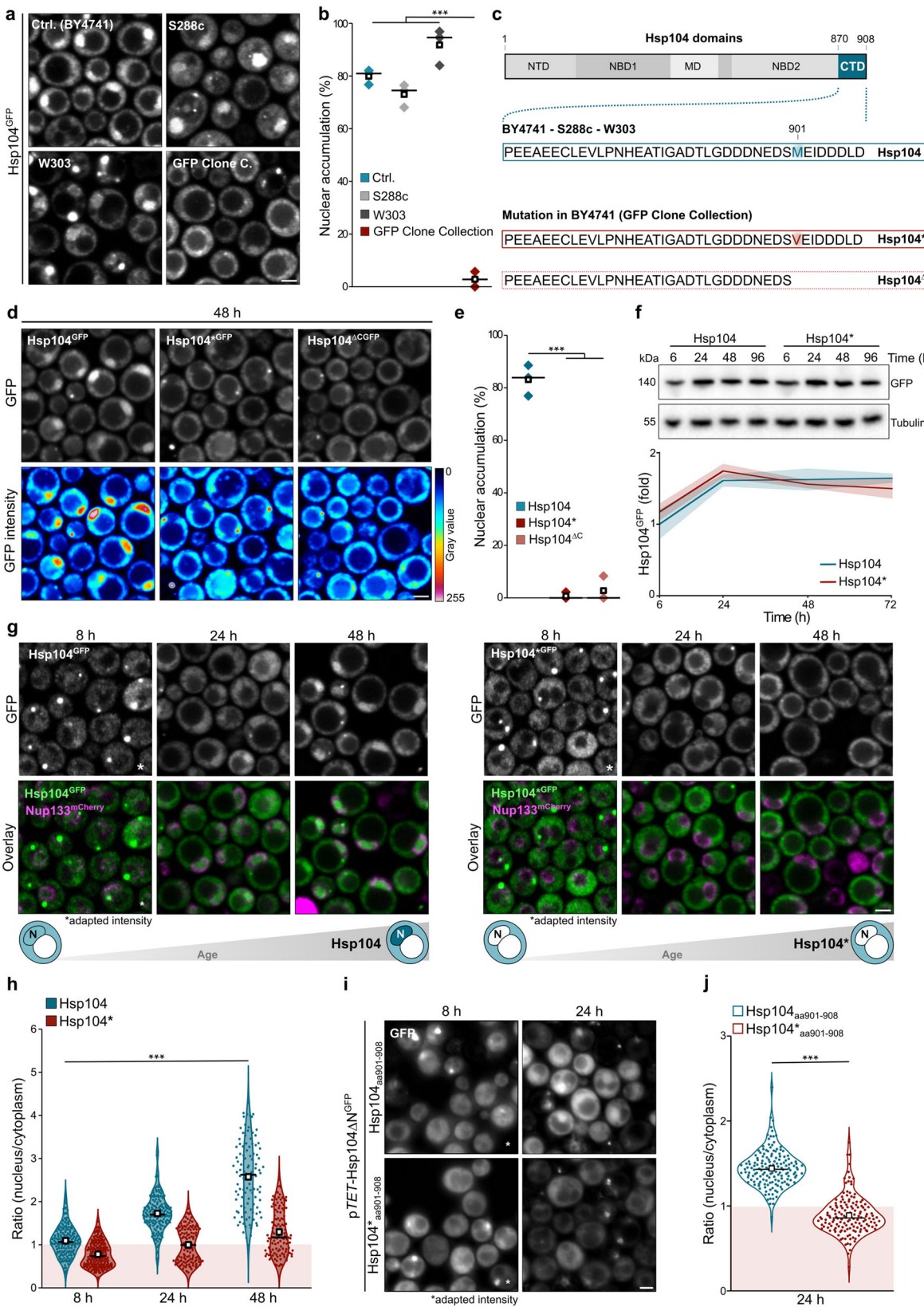

step of translation, we also used lactimidomycin as specific inhibitor of translation initiation[58], which resulted in comparable nuclear Hsp104 targeting (Fig. 6g). Thus, both nuclear entry and exit of Hsp104 are controlled by translation. To assess if nuclear relocalization depends on the ability of Hsp104 to bind its protein substrates, we inhibited the ATPase activity of Hsp104 using guanidine-hydrochloride (Gdn-HCl)[59].

In parallel, we used myricetin and VER-155008 to inhibit Hsp70, which is required for Hsp104 to engage misfolded and aggregated proteins[60,61]. Neither of these substances influenced the nuclear accumulation of Hsp104 upon inhibition of translation (Fig. 6h), demonstrating that the nuclear targeting is independent of substrate engagement and disaggregase activity. These findings reveal that

**Fig. 2 | Nuclear targeting of Hsp104 depends on a motif in its flexible C-terminal tail. a** Micrographs of strains (BY4741, S288c, W303, and BY4741 from the GFP Clone Collection) endogenously expressing Hsp104$^{GFP}$ at 48 h. **b** Quantification of nuclear accumulation of Hsp104$^{GFP}$ in cells described in (**a**). Dot plots with mean (square) and median (line). Each dot represents one biological replicate (Ctrl.: 124 cells; S288c: 166 cells; W303: 94 cells; GFP Clone Collection: 101 cells). **c** Hsp104 sequences and generated mutants (Hsp104* and Hsp104$^{\Delta C}$). N-terminal domain (NTD); Nucleotide-binding domain 1 or 2 (NBD1/2); Middle domain (MD); C-terminal domain (CTD). **d, e** Micrographs (**d**) and corresponding quantification of nuclear accumulation (**e**) of cells harboring endogenously GFP-tagged Hsp104, Hsp104* or Hsp104$^{\Delta C}$ at 48 h. Dot plots with mean (square) and median (line). Each dot represents one biological replicate (Hsp104: 105 cells; Hsp104*: 118 cells; Hsp104$^{\Delta C}$: 122 cells). **f** Immunoblot and quantification of Hsp104$^{GFP}$ levels of strains harboring GFP-chimeras of wild type and mutant

Hsp104. Protein levels were normalized to 6 h wild type Hsp104. Line graph with mean ± s.e.m., *n* = 3 biological replicates. **g, h** Micrographs of cells harboring endogenously GFP-tagged Hsp104 or Hsp104* and Nup133$^{mCherry}$. GFP signal intensity at 8 h was adapted to ensure visibility. **h** Quantification of the GFP intensity ratio between nucleus and cytoplasm of cells described in (**g**). Violin plots with mean (square) and median (line), each dot represents one cell (8 h Hsp104: 166 cells; 8 h Hsp104*: 204 cells; 24 h Hsp104: 166 cells; 24 h Hsp104* 124 cells; 48 h Hsp104: 105 cells; 48 h Hsp104* 118 cells). **i** Micrographs of cells expressing GFP fused to Hsp104$_{aa901-908}$ controlled by the *TET* promoter. Signal intensity at 8 h was increased to ensure visibility. **j** Quantification of the GFP intensity ratio between nucleus and cytoplasm of cells described in (**i**). Violin plots with mean (square) and median (line), each dot represents one cell (Hsp104$_{aa901-908}$: 157 cells; Hsp104*$_{aa901-908}$: 150 cells). Scale bars: 2 μm. ***$p < 0.001$. Source data are provided as a Source Data file. See Supplementary Table 3 for details on statistical analyses.

decreased translation is sufficient to rapidly target Hsp104 to the nucleus even in young cells. Likewise, exit of Hsp104 from the nucleus upon re-feeding of aged cells with glucose is triggered by increased translation.

## Hsp104 manages misfolded proteins in the nucleus of aged cells

We hypothesized that the Hsp104 disaggregase serves a protective function in the nucleus as cells age. First, we asked whether aggregates accumulate in the nucleus during cellular aging. Indeed, transmission electron microscopy showed that electron-dense clusters (EDCs), likely reflecting protein aggregates, appeared in the nuclei of aged but not that of young cells, with 61% of the aged population scoring EDC-positive (Fig. 7a). Importantly, immunogold labeling demonstrated a clear co-localization of native Hsp104 with these clusters (Fig. 7b, Supplementary Fig. 6a for unlabeled micrographs). Likewise, fluorescence microscopy demonstrated the accumulation of nuclear aggregates decorated by Hsp104-GFP upon prolonged aging (Fig. 7c). Hence, cellular aging is accompanied by the emergence of protein aggregates also in the nucleus.

To assess a potential cytoprotective function of nuclear Hsp104, we combined a perturbation of the proteostasis system with the exclusion of Hsp104 from the nucleus. Initially, we established that the age-dependent accumulation of Hsp104 in the nucleus did not impact on the basal transcriptional activity of the heat shock response transcription factor Hsf1 (Supplementary Fig. 6b). We perturbed the proteostasis system by heat shocking cells with nuclear and nuclear-targeting defective Hsp104*. Heat shock induced numerous aggregates decorated by Hsp104 or Hsp104* that over time were disentangled, yet did not affect the overall nuclear *versus* cytoplasmic localization of the respective variant (Supplementary Fig. 6c). Next, we tested whether nuclear Hsp104 has the capacity to reactivate misfolded proteins by using nuclear targeted, temperature-labile firefly luciferase as model substrate ($^{GFP}$FFL-NLS)[62]. We assessed in vivo Hsp104-dependent refolding rates of heat-inactivated firefly luciferase in young and old cells, as measured by the regain of bioluminescence after heat shock. Immediate reactivation of $^{GFP}$FFL-NLS within 30 min after acute heat shock was comparable for Hsp104 and Hsp104* expressing cells. However, specifically in aged cells, wild type Hsp104 reactivated more heat-denatured $^{GFP}$FFL-NLS than the nuclear targeting-defective Hsp104* within the recovery phase (Fig. 7d, e). Thus, initial reactivation of denatured $^{GFP}$FFL-NLS was not impaired by the lack of nuclear Hsp104*, likely due to the dominant reactivation pathways executed by nuclear Hsp70 together with J-domain proteins/Hsp40s (Apj1) and Hsp110s (Sse1/Sse2)[63-65]. Later on, when only the more severely aggregated species of $^{GFP}$FFL-NLS remained, the lower levels of nuclear disaggregation activity in Hsp104* expressing cells became apparent. In line, *hsp104Δ* cells displayed impaired reactivation of $^{GFP}$FFL-NLS immediately after heat shock, likely due to a cell-wide loss of disaggregation activity and thus saturation of the Hsp70-40-110 axis. In aged cells harboring wild type and thus nuclear Hsp104,

the firefly luciferase activity even increased beyond the initial starting levels. This demonstrated that aged cells accumulated a pool of inactive $^{GFP}$FFL-NLS in the nucleus that was reactivated following the heat shock (Fig. 7d, e). To generally enforce the accumulation of misfolded proteins, we impaired proteasomal degradation using MG-132. Of note, this resulted in an increase of protein aggregates specifically in Hsp104* cells that lack nuclear Hsp104 disaggregase activity (Fig. 7f). Nuclear but not cytosolic Hsp104 preferentially interacts with several ubiquitin-proteasome-associated factors, again indicating a role of Hsp104 in the management of nuclear-localized misfolded proteins (Fig. 7g, Supplementary Fig. 5a). Indeed, proteasomal inhibition enforced the targeting of Hsp104 to the nucleus of aged cells (Fig. 7h). Likewise, genetic inactivation of key PQC E3 ubiquitin ligases (*doa10Δubr1Δ*), which remove nuclear and cytosolic misfolded proteins, increased nuclear Hsp104 accumulation in aged cells, suggesting that more misfolded protein species are present in the nucleus in this mutant (Fig. 7i, j). Taken together, these findings support a critical function for Hsp104 in the handling and clearance of nuclear misfolded protein species during cellular aging.

## Nuclear Hsp104 ensures rapid restart of translation to support quiescence exit

Entry into quiescence is accompanied by a reduction in translation, downregulation of the translation apparatus as well as the removal of a large subpopulation of ribosomes via ribophagy[15]. We found that Hsp104 is targeted to the nucleus when translation is downregulated and interacts with eIF2 and a plethora of ribosomal proteins in the nucleus. This suggested that Hsp104 safeguards the translation apparatus during extended aging and entry into the quiescent state. Thus, we tested whether nuclear Hsp104 facilitated the resumption of translation when cells exit quiescence to re-enter the cell cycle. Hsp104 and Hsp104* cells were supplied with fresh nutrients after 1 day or 5 days of aging to trigger regrowth. We followed new protein synthesis by metabolic labeling via $^{35}$S-methionine, which is incorporated into newly synthesized proteins, enabling a clear distinction between nascent species and mature, pre-existing proteins. Interestingly, nuclear but not cytosolic Hsp104 was required for the rapid resumption of translation after an extended period of aging (Fig. 8a). Similar results were obtained using strains with untagged Hsp104 and Hsp104* variants (Supplementary Fig. 7a, b). The restart of protein synthesis was inhibited by treating cells with MG-132 during aging, resembling the defect caused by the lack of Hsp104 in the nucleus (Fig. 8b). This suggests that misfolded proteins damage the functionality of the dormant nuclear translation machinery and supports a function for Hsp104 in the reactivation of misfolded proteins in the nucleus of aged cells, perhaps even involving repair of aberrant translation factors.

Finally, we assessed the function of nuclear Hsp104 in quiescence exit and re-entry into the cell cycle. To this end, cells were aged for 5 days and the accumulation of misfolded proteins was induced via impairment of proteasomal activity using MG-132. We followed

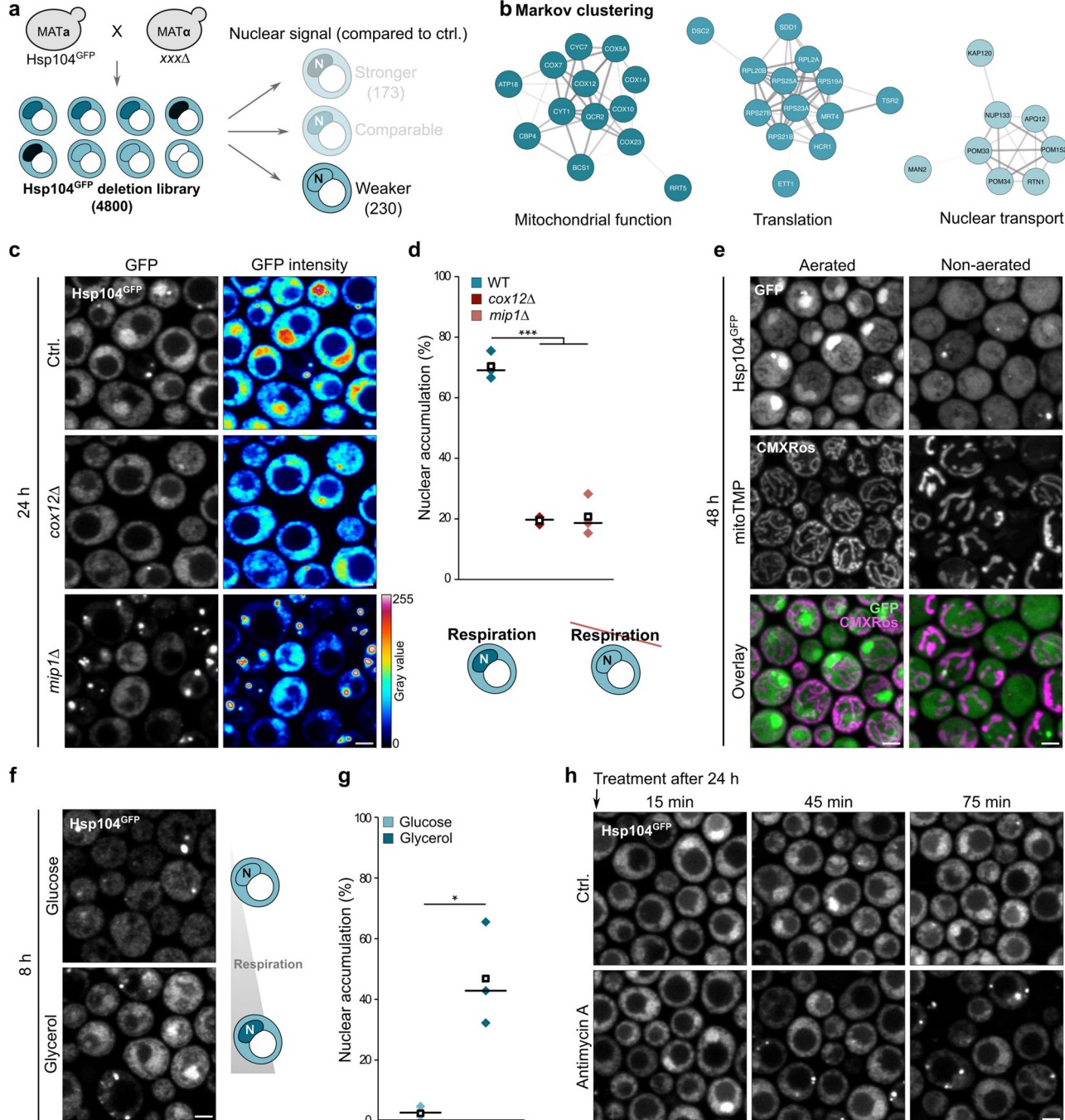

**Fig. 3 | Mitochondrial respiration is critical for nuclear accumulation of Hsp104.** **a** Schematic visualization of generation and analysis of the genome-wide deletion library endogenously equipped with Hsp104[GFP]. Hits scored as "weaker nuclear accumulation" compared to control cells were subjected to further analyses. The number of deletion mutants in respective classes is given in brackets. **b** Visualization of the three most prominent enrichment clusters of hits classified as "weaker nuclear accumulation" compared to control cells after STRING analysis and Markov clustering. See Supplementary Fig. 3a for the complete STRING network. **c** Micrographs of wild type, cox12Δ and mip1Δ cells endogenously expressing Hsp104[GFP] at 24 h. Scale bar: 2 μm. **d** Quantification of nuclear accumulation of Hsp104[GFP] in cells as described in (**c**). Schematic drawings illustrate major phenotypes. N = Nucleus. Dot plots with mean (square) and median (line). Each dot represents one biological replicate (Ctrl.: 249 cells; cox12Δ: 202 cells; mip1Δ: 193

cells). **e** Micrographs of cells endogenously expressing Hsp104[GFP] stained with Mitotracker CMXRos to visualize the mitochondrial transmembrane potential (mitoTMP) at 24 h (maximum intensity projections from Z-stacks). Scale bar: 2 μm. **f** Micrographs of cells endogenously expressing Hsp104[GFP] cultivated in glucose or glycerol-containing media at 8 h. Schematic drawings illustrate major phenotypes. N = Nucleus. Scale bar: 2 μm. **g** Quantification of nuclear accumulation of Hsp104[GFP] in cells as described in (**f**). Dot plots with mean (square) and median (line). Each dot represents one biological replicate (Glucose: 110 cells; Glycerol: 67 cells). **h** Micrographs of cells endogenously expressing Hsp104[GFP] treated with antimycin A to inhibit mitochondrial respiration after 24 h. Scale bar: 2 μm. *$p < 0.05$; ***$p < 0.001$. Source data are provided as a Source Data file. See Supplementary Table 3 for details on statistical analyses.

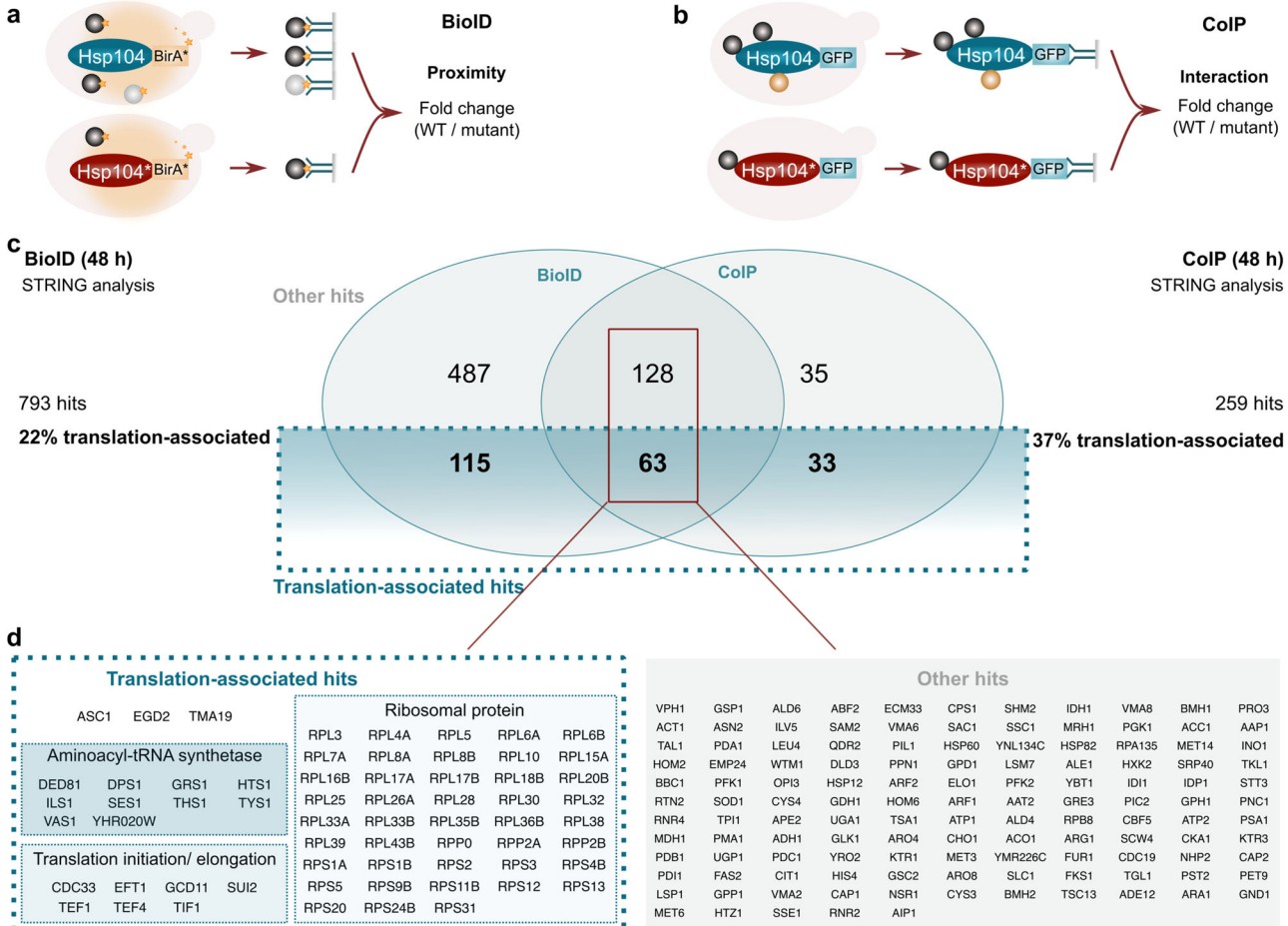

**Fig. 4 | Nuclear Hsp104 interacts with translation-associated factors in aged cells. a** Schematic visualization of the workflow for proximity labeling (BioID) experiments to assess proximity to the bait protein (Hsp104 variants endogenously equipped with the biotin ligase BirA*). **b** Schematic visualization of the workflow for co-immunoprecipitation (CoIP) experiments to assess interaction with the bait protein (Hsp104 variants endogenously equipped with GFP). **c** VENN diagram to visualize hits shared between BioID and CoIP after performing STRING analysis, classified as "Translation-associated" and "Others". **d** Listing of hits from (**c**) present in BioID and CoIP analyses.

regrowth after supplementation of fresh nutrients. MG-132 pre-treatment prominently delayed the resumption of growth, in particular in the absence of nuclear Hsp104 (Fig. 8c and Supplementary Fig. 7c). Interestingly, cells lacking Hsp104 (*hsp104Δ*) displayed a similar growth delay as Hsp104* expressing cells (Fig. 8c). While wild type Hsp104 cells completed the first cell cycle within 9.4 h, *hsp104Δ* cells required 10.3 h for the first doubling and Hsp104* cells needed 10.9 h, suggesting that Hsp104* causes a loss-of-function phenotype in respect to cell cycle re-entry. The subsequent exponential growth rates did not differ between the strains. We ruled out that the delayed resumption of growth was merely the consequence of limiting cytosolic Sui2 levels by conditionally overexpressing Sui2 (Supplementary Fig. 7d, e). Importantly, replenishment of nutrients and thus cell cycle re-entry triggered the rapid nuclear exit of Sui2 only in the presence of nuclear Hsp104 (Fig. 8d, e, Supplementary Fig. 7f). We reasoned that targeting of Sui2 to the nucleus protects it from aggregation. Indeed, perturbing proteostasis by long-term proteasomal impairment (Bortezomib treatment) caused Sui2 to aggregate particularly in cells lacking nuclear Hsp104 (Fig. 8f, g, Supplementary Fig. 7g, h). Upon nutrient replenishment, cells were able to disentangle these aggregates (Fig. 8f, g, Supplementary Fig. 7g, h). Thus, nuclear Hsp104 facilitates efficient nuclear targeting of Sui2 when translational activity decreases in aged cells, protects it from stress-induced aggregation and finally supports nuclear exit of Sui2 upon resumption of translation and growth.

In conclusion, we demonstrate that Hsp104 relocalizes to the nucleus during aging when translation rates drop. During extended aging, the coordinated action of nuclear Hsp104 and the ubiquitin-proteasome system safeguards the dormant translation machinery to enable the resumption of translation and growth when cells exit the quiescent state.

## Discussion

Here we show that the disaggregase Hsp104 is re-directed to the nucleus as cells age, where it conveys a fitness advantage for quiescent cells when they re-enter the cell cycle. Our data indicate that nuclear Hsp104 safeguards the dormant translation machinery to support the rapid resumption of protein biosynthesis upon nutrient replenishment. The cytosolic-nuclear partitioning of Hsp104 is governed by the metabolic status of cells, whereby specifically the output from the translation machinery functions as the regulatory cue. We have identified the nuclear targeting machinery and a critical TPR protein binding motif in the C-terminus of Hsp104 that is required for nuclear entry.

In the nucleus of quiescent cells, Hsp104 decorates and clears protein aggregates and associates with latent translation initiation factors, in particular with eIF2 complexes. Thus, nuclear Hsp104 protects the dormant translation machinery from age-induced damage, enabling rapid resumption of protein synthesis upon quiescence exit. In the exponential phase, cells utilize glucose as a fermentable carbon

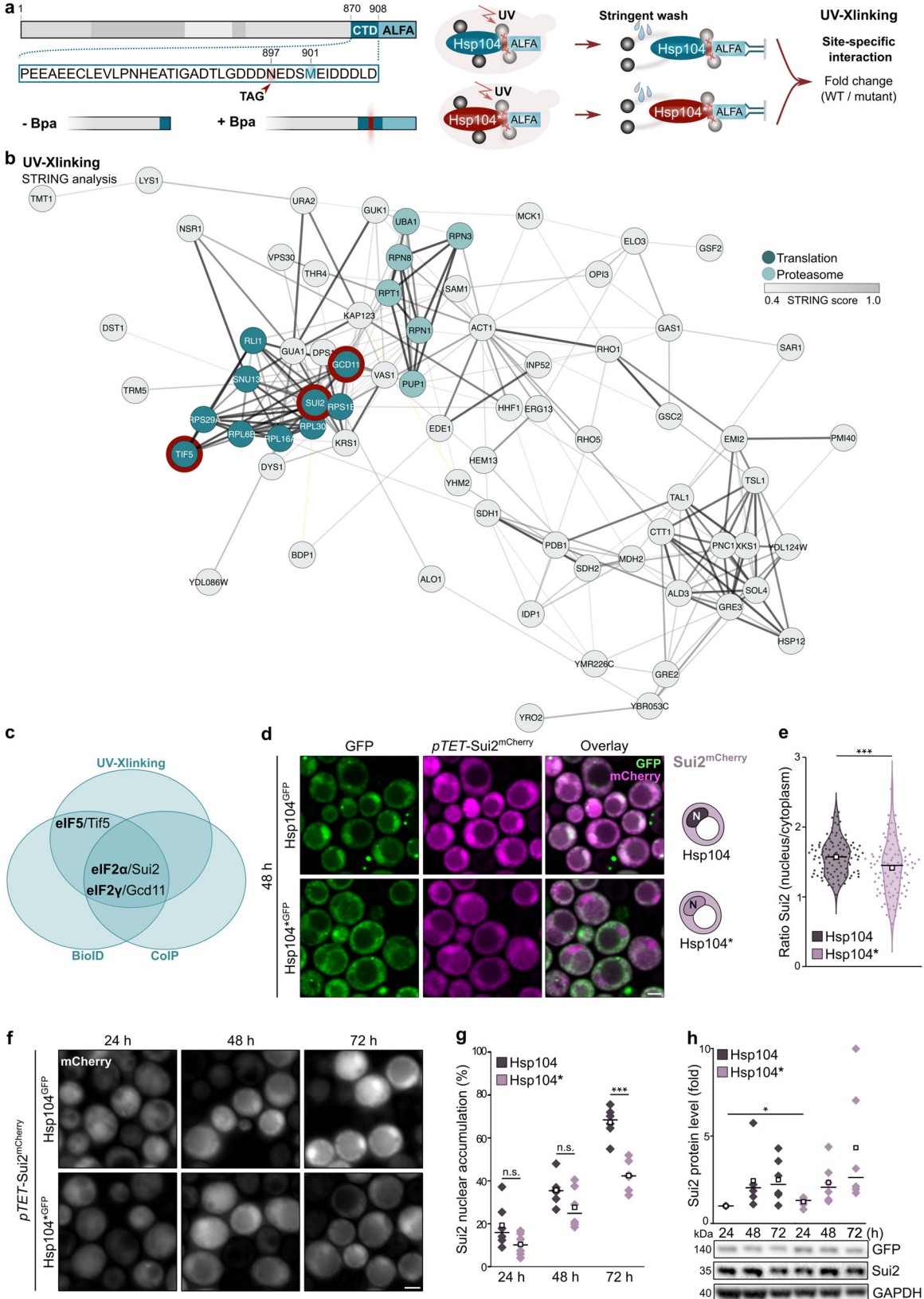

source to rapidly grow and divide, which is accompanied by high translational rates[66]. Nascent proteins are particularly prone to mis-folding and aggregation, creating a need for Hsp104 disaggregation activity in the cytosol[67,68]. As nutrients become limiting, cells switch to respiratory metabolism to prepare for stationary phase and translation rates decrease[56,57,69], reducing the demand for Hsp104 in the cytosol.

Concurrently, cells re-direct Hsp104 to the nucleus, where it associates with and disentangles protein aggregates. Hence, the decreased protein synthesis is likely the most downstream event that drives nuclear targeting of Hsp104. Under our experimental conditions, this function appears to support quiescence exit rather than survival during quiescence, characterized by low translation rates. In line with this notion,

**Fig. 5 | The translation initiation factor eIF2 interacts with nuclear Hsp104.**
**a** Workflow for UV-crosslinking to assess site-specific interaction partners of the bait protein (Hsp104 variants endogenously equipped with the ALFA tag). The UV-crosslinking amino acid p-benzoyl-L-phenylalanine (Bpa) was introduced by stop codon suppression at position 897. **b** STRING network of hits from UV-crosslinking enriched in Hsp104 eluates compared to Hsp104*. Hits associated with "Translation" and "Proteasome" are highlighted and the STRING score is depicted in shades of gray. **c** VENN diagram to visualize the translation initiation factors eIF2α/Sui2, eIF2γ/Gcd11 and eIF5/Tif5 as shared hits between BioID (Fig. 4c, d), CoIP (Fig. 4c, d) and UV-crosslinking (**b**). **d** Micrographs of cells at 48 h harboring endogenously GFP-tagged Hsp104 variants and mCherry-tagged eIF2α/Sui2 controlled by the *TET* promoter in its own locus (p*TET*-Sui2mCherry). See Supplementary Fig. 5d for additional time points. Scale bar: 2 μm. **e** Quantification of Sui2mCherry intensity ratio between the nucleus and cytoplasm of cells described in (**d**). Violin plots with mean

(square) and median (line), each dot represents one cell (Hsp104: 107 cells; Hsp104*: 98 cells). **f** Micrographs of cells expressing Sui2mCherry under the control of a *TET*-promoter in its own locus (p*TET*-Sui2mCherry) at indicated time points. Scale bar: 2 μm. **g** Quantification of nuclear accumulation of Sui2mCherry in cells described in (**f**). Dot plots with mean (square) and median (line). Each dot represents one biological replicate (24 h Hsp104: 789 cells; 24 h Hsp104*: 621 cells; 48 h Hsp104: 401 cells; 48 h Hsp104*: 543 cells; 72 h Hsp104: 294 cells; 72 h Hsp104: 464 cells). **h** Immunoblot and quantification of Sui2 proteins levels in strains harboring GFP-chimeras of wild type and mutant Hsp104. Blots were probed with antibodies directed against GFP, Sui2 and GAPDH. Six biological replicates per strain were assessed. Protein levels were normalized to the 24 h time point of Hsp104 wild type samples. Dot plots with mean (square) and median (line). n.s.: not significant ($p \geq 0.05$); *$p < 0.05$; ***$p < 0.001$. Source data are provided as a Source Data file. See Supplementary Table 3 for details on statistical analyses.

particularly nascent proteins are at risk of misfolding when they inappropriately interact with aggregates[70,71]. Thus, the proteostatic damage that builds up in quiescence when nuclear Hsp104 is missing becomes evident only upon re-start of protein biosynthesis, resulting in inefficient resumption of translation as well as delayed cell cycle re-entry. Furthermore, additional proteasomal inhibition enforces this defect, suggesting that nuclear Hsp104 and the proteasome work in parallel to remove the misfolding protein damage. Along this line, increased levels of Hsp104 have been shown to enhance proteasomal activity in aged cells[72]. We find that also wild type cells accumulate detectable levels of nuclear protein aggregates during aging, likely due to the fact that the rates of Hsp104-dependent disaggregation are lower than the rates of protein aggregation. In sum, Hsp104 is targeted to the nucleus when translation rates drop to work in concert with the proteasome system in the handling of misfolded proteins, ultimately protecting the latent translation machinery.

Nuclear Hsp104 also interacts with structural ribosomal proteins and translation-associated factors, suggesting that these proteins are recognized as misfolded substrates that are repaired by Hsp104-dependent mechanisms. In line, early steps of ribosome assembly take place in the nucleus, and the misfolding of ribosomal proteins ties up Hsp70 chaperone capacity[73–75]. Curiously, nuclear Hsp104 binds latent eIF2, suggesting that the disaggregase plays a direct role in the assembly or maintenance of these critical translation factors during quiescence. When translation rates drop during aging, nuclear Hsp104 facilitates the targeting of eIF2α/Sui2 to the nucleus and protects it from stress-induced aggregation. Upon nutrient replenishment and thus resumption of translation, Hsp104 promotes nuclear exit of Sui2. Hence, the compromised re-start of translation in the absence of nuclear Hsp104 likely reflects a specific involvement in eIF2 mobilization (Fig. 8h).

Nuclear targeting of Hsp104 is regulated by metabolic cues. The switch to respiratory metabolism re-directs Hsp104 to the nucleus. Such metabolic reprogramming is intimately linked to decreased protein synthesis, and we indeed find that genetic and pharmacological inhibition of translation is sufficient to target Hsp104 to the nucleus even in nutrient-rich conditions. Apparently, the translational output is coordinated with the nuclear transport machinery that shuttles Hsp104 to the nucleus. Proteomic profiling identified Hsp104 interactions with nucleoporins and importin β specifically in aged cells. These interactions were absent in the M901V mutant (Hsp104*), highlighting a critical role for the C-terminal TPR protein binding motif to engage the nuclear targeting machinery. This peptide is sufficient to target GFP to the nucleus of specifically aged cells. Thus, we find that the C-terminus of Hsp104 has two distinct functional interactions: (i) association with the nuclear targeting machinery and (ii) direct binding to its eIF2 complex clients. Indeed, many chaperones, including Hsp70, Hsp90 and J-domain proteins, employ their TPR protein binding motifs located to their extreme C-termini for various and complex interactions within the proteostasis network[76–83]. Our finding

that a minimal TPR protein binding motif is required and sufficient for age-dependent nuclear targeting of Hsp104 suggests that this process is controlled by interactions with one or several TPR proteins. TPR proteins have previously been associated with nucleo-cytoplasmic shuttling[50,84–88]. The hexameric conformation of Hsp104 allows for multiple associations with the six flexible C-termini. Nuclear targeting of Hsp104 does not depend on Hsp70 or its own disaggregation activity, ruling out additional piggy-backing on nuclear-targeted mis-folded proteins. Yet, Hsp104 localization appears to be partially regulated by a tug-of-war between its various clients and TPR protein interactions. For instance, blocking mitochondrial respiration and hence mitochondrial protein import, which triggers cytosolic aggregation of mitochondrial precursor proteins[89,90], is sufficient to extract Hsp104 from the nucleus of aged cells.

Cells have evolved distinct mechanisms to ensure that the translational output dynamically matches nutrient availability. As part of these adaptive mechanisms, cells store fundamental factors of the translation machinery for later use, for example in stress granules and P-bodies. Our study implies that also the nucleus serves as a storage site for components of the translation machinery during quiescence. Here, Hsp104 appears to have a key role in protecting these factors from age-induced damage, allowing rapid mobilization once nutrients become available and cells re-enter the cell cycle. Lower translation rates together with a drop in active, cytosolic ribosomes are characteristic for cells approaching quiescence[15,16], thus we propose that dividing cells place Hsp104 in the nucleus before entering quiescence to prepare for future rapid re-activation of the translation apparatus. In sum, we have uncovered a novel spatially-regulated proteostasis mechanism that maintains cellular fitness during aging.

## Methods
### Yeast strains and genetics
Experiments were carried out in BY4741 (MATa, *his3*Δ1, *leu2*Δ0, *met15*Δ0, *ura3*Δ0), except for Fig. 2a, b, where W303 and S288c were used. All yeast strains used in this study are listed in Supplementary Table 1. Yeast transformations were carried out following standard procedures[91]. Gene deletions and endogenous tags were generated via homologous recombination[92,93]. All oligonucleotides are listed in Supplementary Table 2. Notably, at least three transformants were analyzed for all strains to rule out clonogenic variation.

### Culturing conditions and treatments
All strains were grown at 28 °C and 145 rpm shaking on synthetic complete medium (SC), containing 0.17% yeast nitrogen base, 0.5% $(NH_4)_2SO_4$ and 30 mg/l of all amino acids (except 80 mg/l histidine and 200 mg/l leucine), 30 mg/l adenine and 320 mg/l uracil, with 2% glucose (SCD), 2% galactose (SCG) or 3.2% glycerol (SCGly). Overnight cultures were incubated for 16-18 h in SCD and were diluted to $OD_{600}$ 0.1 into baffled Erlenmeyer flasks. Diploid strains were generated by mixing respective BY4741 strains with an untagged BY4742 strain on

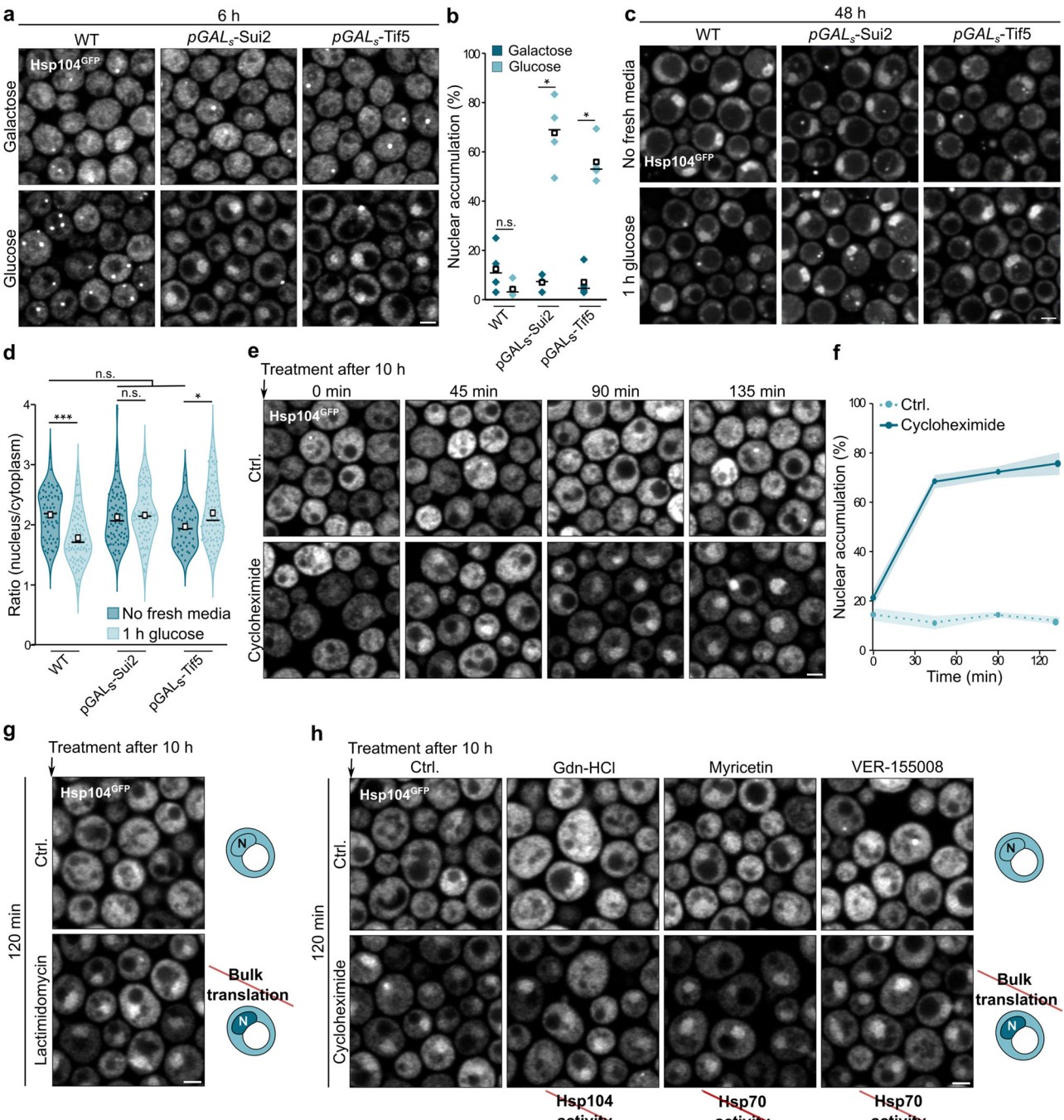

**Fig. 6 | Reduced translation induces nuclear accumulation of Hsp104 independent of its disaggregation activity. a** Micrographs of cells expressing Hsp104[GFP] and eIF2α/Sui2 or eIF5/Tif5 under the control of a galactose-inducible promoter from their native locus at 6 h. Scale bar: 2 μm. **b** Quantification of nuclear accumulation of Hsp104[GFP] in cells described in (**a**). Dot plots with mean (square) and median (line). Each dot represents one biological replicate (Galactose Sui2: 285 cells; Glucose Sui2: 310 cells; Glucose Tif5: 318 cells; Galactose Tif5: 351 cells). **c** Micrographs of cells described in (**a**) at 48 h. Cells were shifted to fresh media 1 h prior to analysis. **d** Quantification of the mean Hsp104[GFP] intensity ratio between the nucleus and cytoplasm of cells described in (**c**). Violin plots with mean (square) and median (line). Each dot represents one cell (No fresh media – WT: 60 cells, p*GAL*s-Sui2: 77 cells, p*GAL*s-Tif5: 45 cells; 1 h Glucose – WT: 83 cells, p*GAL*s-Sui2: 78 cells, p*GAL*s-Tif5: 85 cells). **e** Micrographs of cells endogenously expressing Hsp104[GFP] treated with cycloheximide to inhibit bulk translation after 10 h. Scale bar: 2 μm.

**f** Quantification of nuclear accumulation of Hsp104[GFP] in cells described in (**e**). Line graph with mean ± s.e.m. (0 min Ctrl.: 140 cells; 0 min Cycloheximide: 128 cells; 45 min Ctrl.: 293 cells; 45 min Cycloheximide: 244 cells; 90 min Ctrl.: 303 cells; 90 min Cycloheximide: 253 cells; 135 min Ctrl.: 315 cells; 135 min Cycloheximide: 274 cells). Measurements were taken from micrographs of 3 biological replicates. **g** Micrographs of cells endogenously expressing Hsp104[GFP] treated with lactimidomycin to inhibit bulk translation after 10 h. Schematics illustrate major phenotypes. *N* = Nucleus. Scale bar: 2 μm. **h** Micrographs of cells (*pdr5*Δ) endogenously expressing Hsp104[GFP] pre-treated with guanidine-hydrochloride (Gdn-HCl) to inhibit Hsp104 activity and myricetin or VER-155008 to inhibit Hsp70 activity for 2 h before additional treatment with cycloheximide after 10 h of growth. Schematics illustrate major phenotypes. *N* = Nucleus. Scale bar: 2 μm. Source data are provided as a Source Data file. See Supplementary Table 3 for details on statistical analyses.

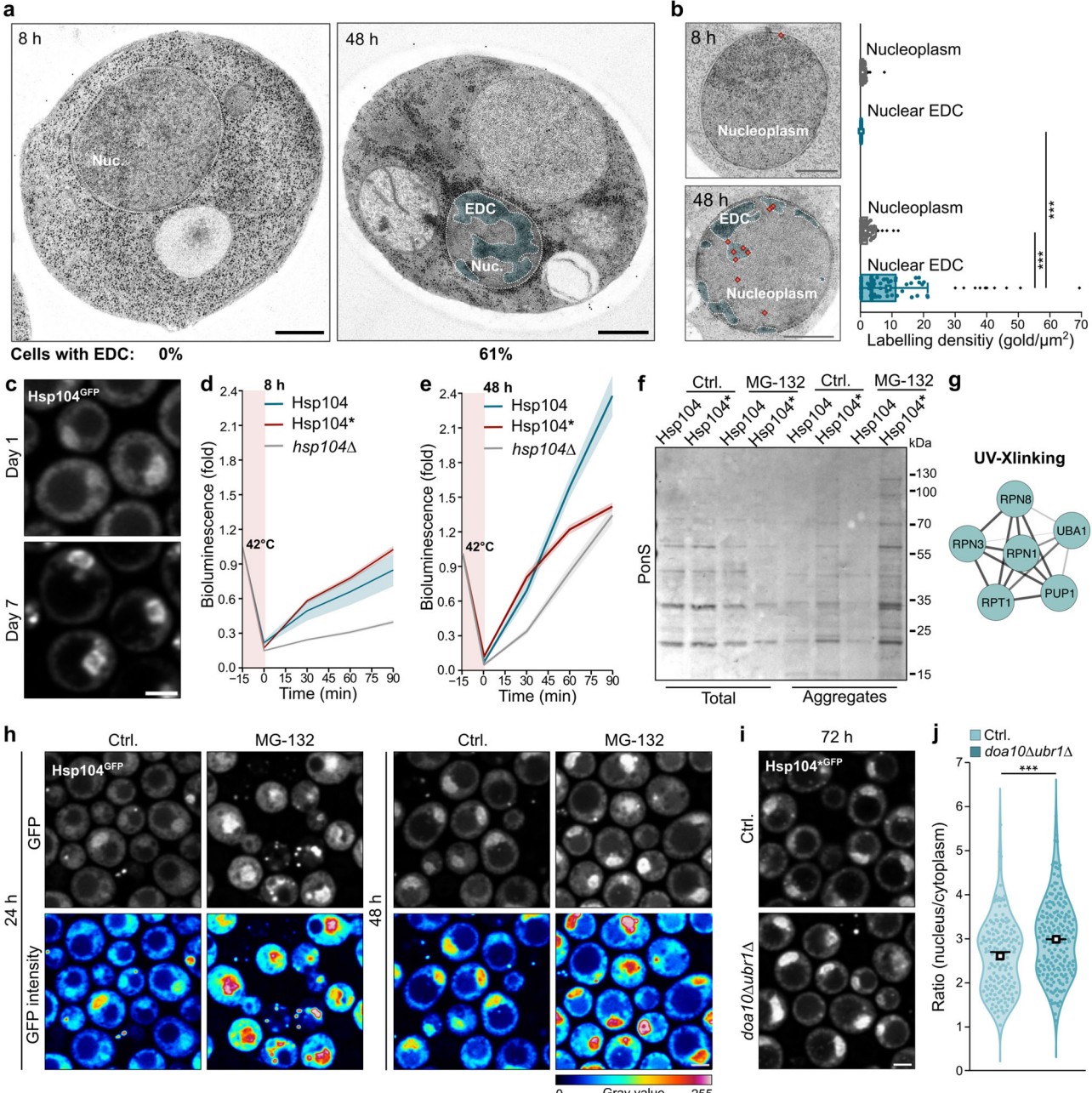

**Fig. 7 | Hsp104 manages misfolded proteins in the nucleus of aged cells.**
**a** Transmission electron micrographs of wild type cells at indicated time points. Nuclear electron-dense clusters (EDC) were quantified (100 cells per time point). Nuc. = Nucleus. Scale bar: 500 nm. **b** Hsp104 immunogold-labeling of cells described in (**a**). Micrographs and quantification of gold labeling density are depicted. EDC and the nucleoplasm are highlighted and the immunogold signal is depicted as diamonds. Box plots with mean (square), median (line) and whiskers with minima and maxima within 1.5 interquartile range are shown (100 cells per time point). Scale bar: 500 μm. **c** Micrographs of cells endogenously expressing Hsp104^GFP on day 1 and day 7. Scale bar: 2 μm. **d, e** Refolding rates of nuclear-targeted, temperature-labile firefly luciferase after heat inactivation at 8 h (**d**) and 48 h (**e**). In vivo refolding was assessed in cells harboring GFP-tagged Hsp104 or Hsp104* as well as *hsp104Δ* cells. Line graph with mean ± s.e.m., *n* = 4 biological replicates. **f** Ponceau S (PonS) stained membranes to visualize proteins from total cellular lysates (left) and isolated aggregates (right) of cells

(*pdr5Δ*) harboring GFP-tagged Hsp104 or Hsp104* on day 5. Cells were treated with MG-132 at inoculation for proteasomal inhibition. For the aggregate fractions, correspondingly 18x more sample was loaded to ensure visibility.
**g** Visualization of genes associated with the term "Proteasome" as prominent enrichment cluster from UV-crosslinking experiments. See Fig. 5b for the complete STRING network and Supplementary Fig. 5a for enrichment analysis via Markov clustering. **h** Micrographs of cells (*pdr5Δ*) harboring Hsp104^GFP treated with MG-132. Scale bar: 2 μm. **i** Micrographs of cells (wild type and *doa10Δubr1Δ*) endogenously expressing Hsp104^GFP at 72 h. Scale bar: 2 μm. **j** Quantification of the mean GFP intensity ratio between the nucleus and cytoplasm of cells described in (**i**). Violin plots with mean (square) and median (line). Each dot represents one cell (Ctrl.: 157 cells; *doa10Δubr1Δ*: 182 cells). Measurements were taken from 3 biological replicates. ***p < 0.001. Source data are provided as a Source Data file. See Supplementary Table 3 for details on statistical analyses.

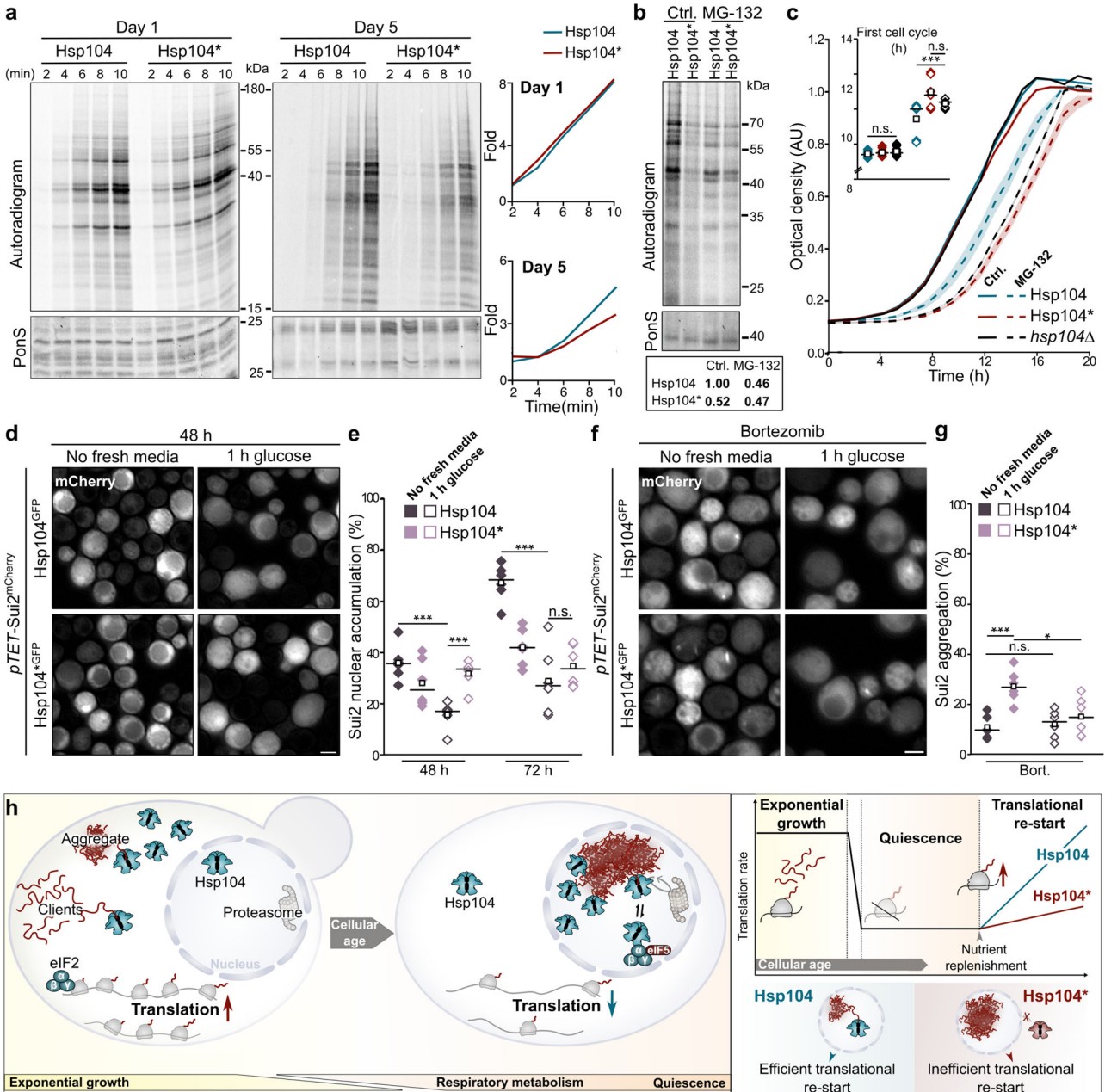

**Fig. 8 | Nuclear Hsp104 ensures rapid restart of translation to support quiescence exit. a** [$^{35}$S]-methionine radiolabeling of nascent proteins in cells harboring endogenously GFP-tagged Hsp104 or Hsp104*. Quantification of the autoradiogram (line graphs), with ponceau S (PonS) as loading control. **b** [$^{35}$S]-methionine radiolabeling of nascent proteins from cells (*pdr5Δ*) harboring endogenously GFP-tagged Hsp104 or Hsp104* on day 5. MG-132 was added at inoculation. PonS served as loading control. Relative autoradiogram intensity is provided. **c** Regrowth kinetics of cells (*pdr5Δ*) harboring endogenously GFP-tagged Hsp104 or Hsp104* as well as of cells lacking Hsp104 cultured for 5 days. MG-132 was added at inoculation. Time for completion of the first cell cycle (inlet). Line graph with mean ± s.e.m., *n* = 9 biological replicates. **d** Micrographs of cells expressing mCherry-tagged eIF2α/Sui2 controlled by the *TET* promoter (p*TET*-Sui2^mCherry). At 48 h cells were shifted to fresh media 1 h prior to analysis. Control conditions overlap with Fig. 5f, g. Scale bar: 2 μm. **e** Quantification of nuclear accumulation of

Sui2 in cells described in (**d**) and cells shifted to fresh media at 72 h (corresponding micrographs in Supplementary Fig. 7f). Dot plots with mean (square) and median (line). Each dot represents one biological replicate (Ctrl − 48 h Hsp104: 401 cells, 48 h Hsp104*: 543 cells; Glucose − 48 h Hsp104: 436 cells, 48 h Hsp104*: 340 cells; Ctrl − 72 h Hsp104: 294 cells, 72 h Hsp104*: 464 cells; Glucose − 72 h Hsp104: 406 cells, 72 h Hsp104*: 378 cells). **f** Micrographs of cells described in (**d**) but with the addition of Bortezomib at initial inoculation. **g** Quantification of aggregation of Sui2 in cells described in (**d**). Dot plots with mean (square) and median (line). Each dot represents one biological replicate (Bort. Ctrl − 48 h Hsp104: 517 cells, 48 h Hsp104*: 523 cells; Bort. Glucose − 48 h Hsp104: 465 cells, 48 h Hsp104*: 442 cells). **h** Model describing Hsp104 function and regulation. n.s.: not significant (*p* ≥ 0.05); **p* < 0.05; ***\**p* < 0.001. Source data are provided as a Source Data file. See Supplementary Table 3 for details on statistical analyses.

full medium (YPD; 1% yeast extract, 2% peptone and 4% glucose) without selection. After incubation for 1 day at 30 °C, cells were re-streaked on respective media to select for diploids. For assessment of Hsp104^GFP localization in aerated and non-aerated conditions (Fig. 3e), cells were inoculated into 96-deepwell plates. For aerated conditions,

cells were cultivated for 48 h at 28 °C with 1,000 rpm shaking to ensure oxygen availability, and for non-aerated conditions cells were incubated at 28 °C without shaking. To inhibit respiration, antimycin A (50 μM final concentration, stock solubilized in ethanol, Sigma-Aldrich) was added to the culture after 24 h of growth before subjecting cells to

microscopic analysis at indicated time points after treatment. Inhibition of bulk translation was achieved by the addition of cycloheximide (1 mg/l final concentration, Sigma-Aldrich) to wild type cells or lactimidomycin (3 μM final concentration, stock solubilized in DMSO, Millipore) to *pdr5Δ* cells. Hsp70 inhibition was achieved by pre-treating *pdr5Δ* cells with VER-155008 (50 μM final concentration, stock solubilized in DMSO, Sigma-Aldrich) or myricetin (100 μM final concentration, solubilized in DMSO, Sigma-Aldrich) for 2 h before microscopic analysis at indicated time points. Similarly, Hsp104 activity was inhibited by treating cells with guanidine-hydrochloride (Gdn-HCl, 5 mM final concentration, Sigma-Aldrich) for 2 h before microscopic analysis. For proteasomal inhibition, MG-132 (50 μM final concentration, solubilized in DMSO, Enzo life sciences) was added to *pdr5Δ* cells or Bortezomib (1 mM final concentration, solubilized in DMSO, Sigma-Aldrich) at inoculation. The respective volumes of ethanol or DMSO were used as solvent control and added to control conditions (Ctrl.).

## Microscopy

For microscopic analysis, cells were harvested at indicated time points and immobilized on agar pads (3% agarose in PBS (25 mM potassium phosphate, 0.9% NaCl; adjusted to pH 7.2)/appropriate medium). To visualize the mitochondrial transmembrane potential, cells corresponding to $OD_{600}$ 1 were harvested and resuspended in 500 μl Mitotracker Red CMXRos (200 nM final concentration; stock solubilized in DMSO, Thermo Fisher) in PBS. After 10 min incubation in the dark, cells were washed in PBS before microscopic analysis. Cells were visualized using a Zeiss LSM 700 confocal microscope equipped with a Plan-Apochromat 63× NA 1.4 DIC M27 oil immersion objective using appropriate settings or a Zeiss Axioskop epifluorescence microscope (Figs. 5f, 8d, f, Supplementary Fig. 7f, g). Z-stacks (Fig. 3e) were acquired using 64x64x12.6 (x/y/z) nm sampling (using the ZEN blue software) and illustration was done by using the maximum-intensity projection method. Micrographs were analyzed and processed with the open-source software Fiji[94]. Gaussian filtering (σ = 0.9-1.1 and xσ = yσ = zσ = 1 for Z-stacks) was applied to decrease image noise, followed by background subtraction (rolling ball radius = 100 pixels). Widefield micrographs in Fig. 8d, f and Supplementary Fig. 7f, g were additionally subjected to the "Unsharp-Mask" Tool (Radius 1; Mask Weight 0.7). For visualization purposes, the dynamic range of the presented micrographs was adapted using the "Brightness/contrast" tool of Fiji, respective lookup tables were assigned, and representative segments were clipped to present higher magnifications. Of note, micrographs within one time point and/or experiment were captured and processed identically. In case the brightness of selected micrographs was enhanced to ensure better visibility, this is specified in the respective figure legend and highlighted accordingly (*).

## Quantification of micrographs

To quantify the portion of cells showing nuclear accumulation of Hsp104$^{GFP}$ or Sui2$^{mCherry}$, cells from micrographs corresponding to at least three biological replicates were analyzed and the percentages of cells with nuclear accumulation were visualized as dot blots (each dot corresponds to one biological replicate). The total number of quantified cells per condition/time point is provided in respective figure legends. To quantify the extent of nuclear accumulation of Hsp104$^{GFP}$ or Sui2$^{mCherry}$, the freehand tool of Fiji was applied to define nuclear and cellular borders, respectively. The nuclear mean GFP/mCherry intensity was subtracted from the mean GFP/mCherry signal of the whole cell to retrieve the mean GFP/mCherry signal intensity of the cytoplasm. Finally, the ratio between nuclear and cytoplasmic mean GFP/mCherry signal intensity per cell was plotted as a single dot in a violin plot. Of note, at least three biological replicates were assessed per condition/time point and the total number of quantified cells per condition/time point is provided in respective figure legends.

## Transmission electron microscopy

Untagged wild type cells were grown for 8 h or 48 h in SCD media at conditions described above. The sample preparation for transmission electron microscopy was performed as described in[95]. In short, cells were filtered from the culture medium and scraped off the filter membrane, loaded into the high-pressure freezing carrier and high-pressure frozen using a Wohlwend Compact 03 (M. Wohlwend GmbH). Cells were freeze substituted in 2% uranyl acetate dissolved in 10% methanol and 90% acetone at −90 °C for 1 h inside a Leica EM AFS2 (Leica Microsystems). Samples were washed by two changes of dehydrated acetone while the temperature was increased to −50 °C overnight. Samples were infiltrated with increasing amounts of Lowicryl HM20 (Polysciences) mixed with acetone (1:4, 2:3, 1:1, 4:1 and 100% 3x) at −50 °C, each step lasting 2 h. The resin was then polymerized using UV light for 72 h at −50 °C, followed by 24 h at room temperature. Ultrathin sections of the yeast cells embedded in plastic were cut with an ultra 45° diamond knife (Diatome) mounted on a Reichert-Jung Ultracut E Ultramicrotome (C. Reichert). Immunolabeling was performed using 1:100 dilutions of anti-Hsp104 (Abcam, ab69549) primary antibodies in block buffer. Primary antibodies were incubated overnight at 4 °C after fixation in 1% paraformaldehyde for 10 min and blocked in 0.8% bovine serum albumin in PBS with 0.1% fish skin gelatine for 1 h. The secondary antibody, 10 nm gold-labeled Goat-anti-Rabbit IgG (H&L) (Electron Microscopy Sciences, Cat#25108), was applied at dilutions of 1:20 for 1 h at room temperature, followed by fixation in 2.5% glutaraldehyde in dH₂O. Samples were on-section stained using 2% aqueous uranyl acetate for 5 min followed by Reynold's lead citrate for 1 min. Imaging was performed on a FEI Tecnai G2 Spirit operated at 120 kV equipped with an Ceta 16 M camera (4k × 4k) (Thermo Fisher Scientific). Sections were counted for quantification of the portion of cells with and without nuclear electron-dense clusters (EDC). For quantification of the labeling density (gold/μm²) in Fig. 1g and Fig. 7b, the respective compartments were encircled using IMOD[96], gold beads were manually identified and the resulting models were exported as a wimp file. Mitochondria were used as negative control. For clarity, selected compartments were highlighted using transparent colors and gold beads were visualized as red diamonds.

## [³⁵S]-methionine radiolabeling of nascent proteins

Cells were harvested at indicated time points, washed three times in ultra-pure H₂O and resuspended in 2 ml medium without amino acids and carbon source. Cells corresponding to $OD_{600}$ 4 were collected via centrifugation and resuspended in 1 ml medium containing 2% glucose and 12.13 μg/ml of all amino acids, except 66.67 μM cysteine and no methionine. 22 μCi [³⁵S]-methionine was added to start the labeling reaction. Pulse-labeling was performed for indicated time intervals (Fig. 8a: every 2 min for 10 min; Fig. 8b: 10 min, Supplementary Fig. 7a, b: 10 min) at 30 °C, 600 rpm and 200 μl of each sample were collected in a fresh tube containing 50 μl of pre-cooled stop solution (1.85 M NaOH, 1 M 2-mercaptoethanol, 20 μM phenylmethylsulfonyl fluoride (PMSF), 33 mM non-radioactive methionine). Samples were incubated for 10 min on ice and tri-chloro acetic acid (TCA) was added to a final concentration of 14%. After an additional incubation time of 30 min on ice, samples were centrifuged for 30 min at 20,000 g and 4 °C. The pellets were rinsed in 1 ml acetone, centrifuged again, resuspended in 75 μl 1x Laemmli buffer (100 mM Tris-HCl pH 6.8, 4% SDS, 20% glycerol, 0.2% bromophenol blue, 100 mM dithiothreitol (DTT)) and boiled at 95 °C for 10 min. 20 μl of each sample was applied for SDS-PAGE analysis. Immunoblotting on nitrocellulose membranes was performed for 3 h at 4 °C. Subsequent Ponceau S staining of the membrane served as loading control. The radioactivity of the membrane was assessed using a RadEye™ B20 Geiger counter (Thermo Scientific). Membranes were incubated with storage phosphor screens in X-ray film cassettes and detected in a FLA 9,000 radioisotope scanner (FujiFilm). Visualization and quantification of the signal were performed using Fiji.

## Isolation of protein aggregates

Cells corresponding to $OD_{600}$ 50 were harvested, washed once with double-distilled water and resuspended in 1 ml lysis buffer (100 mM Tris·HCl pH 7.5, 100 mM NaCl, 5 mM EDTA, 1 mM DTT, 1 mM PMSF, 5% glycerol, 20 mM N-ethylmaleimide). Appr. 500 μl of glass beads (0.5 mm diameter) were added and mechanical lysis was performed in three cycles of 1 min. Samples were cleared from unbroken cells and debris by centrifugation for 5 min at 3500 g at 4 °C. Protein concentration was determined with Bradford assay (Bio-Rad) and samples were adjusted to the same protein concentration (2 mg/ml) with lysis buffer. To assess the whole cell lysate, 100 μl of cleared lysate was mixed with the same volume of 2x Laemmli buffer and incubated for 15 min at 95 °C, before proceeding with immunoblotting procedures as described below. To assess aggregates, 1.5 ml of the cleared lysate (2 mg/ml) was centrifuged for 20 min and 20,000 g at 4 °C. The resulting pellet was resuspended in 300 μl of lysis buffer with 2% IGEPAL CA-630 (Sigma-Aldrich) and sonicated for 5 s (60% amplitude), followed by a second centrifugation step. The pellet was resuspended in 70 μl of 1x Laemmli buffer and the samples were prepared as described for the whole cell lysate.

## Protein expression analysis via immunoblotting

For expression analyses, cells were harvested at indicated time points and resuspended in pre-cooled lysis buffer (1.85 M NaOH, 1 M 2-mercaptoethanol, 20 μM PMSF). After a 10 min incubation on ice, TCA solution was admixed (final concentration of 14%) and samples were incubated for 30 min on ice. The cell extract was pelleted for 15 min at 20,000 g and 4 °C. The pellet was washed once in ice-cold acetone and the centrifugation step was repeated. The supernatant was removed and the pellet was resuspended in Laemmli buffer. After incubation for 10 min at 65 °C, samples were loaded on SDS-acrylamide gels using Tris-glycine running buffer (25 mM Tris base; 200 mM glycine; 0.05% SDS). Proteins were separated by electrophoresis and blotted onto PVDF membranes using wet electro-transfer protocols. After blocking in 5% milk powder solubilized in TBS (500 mM Tris-HCl pH 7.4, 1.5 M NaCl), membranes were decorated with primary antibodies against the GFP-epitope (dilution 1:2500, mouse, Sigma-Aldrich), the ALFA-epitope (dilution 1:10,000, rabbit, NanoTag Biotechnologies), tubulin (dilution 1:10,000, rabbit, Abcam) and GAPDH (dilution 1:10,000, rabbit, Thermo Fisher Scientific). Subsequently, blots were probed with respective peroxidase-conjugated secondary antibodies against mouse (dilution 1:10,000, rabbit, Sigma-Aldrich) or rabbit (dilution 1:10,000, goat, Sigma-Aldrich). Clarity Western ECL Substrate (Bio-Rad) was used for detection on a ChemiDocXRS+Imaging System (Bio-Rad) and densitometric quantification was performed with Image Lab 5.2.1 Software (Bio-Rad). Signals were normalized to the respective loading control and fold changes (specified in the corresponding figure legend) were calculated. Uncropped blots are provided in the Source Data.

## HSP104^GFP single deletion library construction and imaging

The library was constructed using the synthetic genetic array (SGA) technology[97,98]. The query strain harboring HSP104-GFP:LEU2 was introduced into the library by mating it with the SGA-v2 single gene deletion library. Diploids were selected using SC-Leu + G418 (Fisher Scientific CAS No. 108321-42-2) and cells were transferred to sporulation plates and incubated for 14 days. Haploids were selected using SC-His-Arg-Lys-Leu + L-canavanine (Sigma Aldrich, CAS No.543-38-4) + S-(2-Aminoethyl)-L-cysteine hydrochloride (Sigma Aldrich, CAS No.4099-35-8). For the final selection, additionally G418 was added.

For the microscopic analysis, clones were inoculated into 384-midi-well plates filled with SCD medium with appropriate selection. After 48 h of incubation (1000 rpm at 28 °C), 50 μl of cells were transferred into glass-bottom 384-well plates (Cellvis, P384-1.5H-N) that were pre-treated with Concanavalin A. After 5 min of incubation, cells were washed twice before microscopic analysis using a Zeiss Axio Observer 7 widefield microscope (Plan-Neofluar 100x/1.3 oil; Filter set: Zeiss set 10 - FITC (450-490; 510; 515-565)) and 4 images per strain were captured. Image processing was performed using Fiji[94] and phenotypes were compared to wild type micrographs. After classification, hits scored as "weaker nuclear accumulation" compared to control cells were subjected to STRING analyses and Markov clustering to identify enrichment clusters. Cytoscape (v. 3.9.1) was used for analyses and network visualization[99].

## Analysis of cell growth

Cellular regrowth was analyzed with a Bioscreen CTM automated microbiology growth curve analysis system (Growth Curves, USA) using the BioScreen CTM Software. Pre-cultures were prepared as described above and MG-132 was added for proteasomal inhibition. After culturing for 5 days, cells were washed twice and re-inoculated into fresh SCD medium to $OD_{600}$ 0.1 in the supplier's "honeycomb microplates". Thereby, a final volume of 250 μl per well was used and the $OD_{600}$ was automatically measured at 28 °C and shaking at the maximum level. The respective medium without cells was used as blank. The time until the first cell cycle was completed (doubling of initial absorbance value) was assessed for each individual biological replicate and visualized as box plot inserted into the main graph.

## Hsp104-dependent refolding of firefly luciferase

To assess in vivo refolding rates of nuclear-targeted, temperature-labile firefly luciferase (FFL) after heat inactivation, strains harboring FFL-NES were grown at 28 °C for 8 h or 48 h. Prior to the 15 min heat shock at 42 °C, cells were treated with cycloheximide (100 mg/l final concentration) to inhibit the new synthesis of FFL and the first sample was collected for measurement. Further samples were taken every 15 min after the heat shock. D-luciferin (in SCD medium, Sigma-Aldrich) was admixed to the collected samples at a final concentration of 455 mg/l in white-bottom low-well plates and luminescence was assessed with an Orion II Microplate Luminometer (Berthold Detection Systems GmbH). The initial bioluminescence values were used as a reference for fold determination. 4 biological replicates per condition were assessed and hsp104Δ was used as control.

## Assessment of the heat shock response and the general stress response via Nanoluc

These assays were performed as previously described using the Nano-Glo Luciferase assay system (Promega)[100]. 30 μl of cells were harvested into white-bottom low-well plates at indicated time points and 3 μl substrate-buffer mix (Nano-Glo substrate, diluted 10-fold in buffer supplied with the kit) were admixed. After 5 min of incubation, luminescence was assessed with an Orion II Microplate Luminometer (Berthold Detection Systems GmbH) using the Orion II Microplate Luminometer Software.

## Proximity-labeling with BioID

Cells harboring endogenously BirA*-tagged Hsp104 variants were inoculated to $OD_{600}$ 0.1 and biotin (50 μM final concentration, stock solubilized in DMSO) was added 9 h before harvesting. 48 h after inoculation, samples were collected by centrifugation (5 min, 4,000 g) and washed once in water with 10 mM sodium azide. Cell pellets were resuspended in 500 μl ice-cold RIPA buffer without detergents (50 mM Tris-HCl pH 7.5, 150 mM NaCl, 1.5 mM $MgCl_2$, 1 mM EGTA, 1 mM DTT, 1 mM PMSF, Roche cOmplete EDTA-free protease inhibitor cocktail). Appr. 500 μl of glass beads (0.5 mm diameter) were added and mechanical lysis was performed in three cycles of 1 min. Following cell lysis, 0.4% SDS and 1% (v/v) IGEPAL CA-630 (Sigma-Aldrich) were admixed and samples were sonicated for 3 cycles of 10 s (60% amplitude). After incubation for 20 min at room temperature, cell debris and insoluble material were removed by centrifugation (24,000 g, 15 min, 4 °C). The cleared lysates were transferred to fresh tubes containing 30 μl pre-equilibrated Pierce™ Streptavidin Magnetic Beads (60 μl

slurry, Thermo Scientific). After 3 h of incubation at 4 °C tumbling, the beads were washed extensively with RIPA buffer (with 0.1% SDS), followed by 3x washes with 50 mM ammonium bicarbonate buffer. Finally, 200 μl of ammonium bicarbonate buffer and sequencing-grade trypsin (1 μg, Promega) were added to the beads. After overnight digestion at 37 °C, the generated peptides were lyophilized and analyzed by mass spectrometry. The LC-MS/MS workflow is described in detail in[101]. The codes to perform analyses on the BioID LC-MS data are available at GitHub repository (https://github.com/wasimaftab/LIMMA-pipeline-proteomics/tree/master).

## Co-immunoprecipitation

Cells were inoculated to $OD_{600}$ 0.1 into 200 ml SCD medium and incubated for 48 h before harvesting 2,000 $OD_{600}$ by centrifugation (5 min, 4,000 g). The cell pellet was washed once with distilled water and resuspended in 15 ml ice-cold lysis buffer (100 mM Tris-HCl pH 6.8, 150 mM NaCl, 1 mM EDTA, Roche cOmplete EDTA-free protease inhibitor cocktail). Homogenization was conducted via an Avestin Emulsiflex C-15, applying a homogenization pressure of 18,000 psi. Samples were cleared from cell debris and non-lysed cells by centrifugation for 8 min at 1000 g and 4 °C. Protein concentration was determined with Bradford assay (Bio-Rad). 2 mg of total protein were transferred to low-bind tubes and IGEPAL CA-630 (1% (v/v) final concentration, Sigma-Aldrich) was added. After incubation for 30 min on ice, cell debris and insoluble material were removed by centrifugation (24,000 g, 15 min, 4 °C). The cleared lysates were transferred to fresh low-bind tubes containing 30 μl pre-equilibrated GFP-trap Magnetic Agarose (60 μl slurry, Chromotek). After incubation for 2 h at 4 °C tumbling, the supernatant was removed and samples were washed 5x with wash buffer (100 mM Tris-HCl pH 6.8, 150 mM NaCl, 1 mM EDTA, 0.1% IGEPAL CA-630). Finally, the supernatant was removed, beads were flash-frozen and stored at −80 °C until further processing. Downstream processing and mass spectrometry is described below.

## UV-activated site-specific crosslinking

Cells were inoculated to $OD_{600}$ 0.1 in duplicates into 50 ml of respective selection media (SCD without tryptophan). After 4 h, p-benzoyl-L-phenylalanine (Bpa, 1 mM final concentration, solubilized in 1 M NaOH, Fisher Scientific) and respective solvent controls were added to the cultures, which were harvested by centrifugation after 24 h. Each cell pellet was resuspended in ice-cold water, half of the sample was pelleted and flash frozen (control), while the other half was UV-crosslinked on ice in 6-well plates using a BL350 UV-A lamp and subsequently flash frozen. The pellets were thawed on ice and resuspended in 1 ml of 8 M urea lysis buffer (8 M urea, 100 mM Tris-HCl pH 7.4, 150 mM NaCl, 1 mM EDTA pH 8, Roche cOmplete EDTA-free protease inhibitor cocktail). Appr. 500 μl of glass beads (0.5 mm diameter) were added and mechanical lysis was performed in three cycles of 1 min at maximum velocity. Protein concentration was determined with Bradford assay (Bio-Rad) and samples were adjusted to the same protein concentration. Samples were incubated for 30 min at room temperature tumbling, followed by a 15 min centrifugation step at 24,000 g to remove debris and insoluble material. 800 μl of cleared lysate was mixed with 800 μl lysis buffer without urea (100 mM Tris-HCl pH 7.4, 150 mM NaCl, 1 mM EDTA pH 8) and transferred to a fresh tube containing 20 μl pre-equilibrated ALFA Selector^ST beads (40 μl slurry, NanoTag Biotechnologies). After 2 h incubation at room temperature tumbling, the suspension was transferred to pre-equilibrated spin columns (Micro Bio-Spin columns, Bio-Rad) and the supernatant was removed by centrifugation (30 s, 1,000 g). The samples were washed 1x with 4 M urea lysis buffer (4 M urea, 100 mM Tris-HCl pH 7.4, 150 mM NaCl, 1 mM EDTA), 3x with wash buffer 1 (100 mM Tris-HCl pH 7.4, 0.1% SDS), 3x with wash buffer 2 (8 M Urea in 100 mM Tris untitrated) and 3x with elution buffer (50 mM ammonium bicarbonate, pH 8). Finally, the samples were resuspended in 2 × 500 μl elution buffer

and transferred to a fresh tube, where the supernatant was removed and the protein-bound beads were flash frozen, ready for downstream processes. Downstream processing and mass spectrometry is described below.

## Downstream processing of proximity labeling and UV-crosslinking experiments

On-bead reduction, alkylation and digestion (trypsin, sequencing grade modified, Pierce) was performed followed by SP3 peptide clean-up of the resulting supernatant[102]. Each sample was separated using a Thermo Scientific Dionex nano LC-system in a 3 hr 5-40% ACN gradient coupled to Thermo Scientific High Field QExactive. All searches were done against the *Saccharomyces cerevisiae* Uniprot database (accessed in 20220227) using our proteomics workflow (https://github.com/lehtiolab/ddamsproteomics vs2.7). Protein false discovery rates were calculated using the picked-FDR method using gene symbols as protein groups and limited to 1% FDR[103].

## Network analyses of hits

Pre-processed data from Co-IP and UV-crosslinking experiments were analyzed according to recommendations from the Clinical Proteomics Mass Spectrometry at Karolinska University Hospital and Science for Life Laboratory. Briefly, only proteins were considered that emerged in at least two out of three replicates and were above the background signal (untagged control). Then, the average from the peak area was calculated to get the fold change between Hsp104 and Hsp104*. Hits enriched in Hsp104 eluates from BioID, CoIP and UV-crosslinking experiments were subjected to network analyses using Cytoscape (v.3.9.1), either using fold change values for visualization or subjecting genes to STRING analyses and Markov algorithm calculations to identify statistically significant enrichment clusters[99]. STRING parameters were used to group hits into "Translation" and "Proteasome". Further, detailed analyses on proteins associated with ribosome (biogenesis) and/or translation were performed by extracting GO terms/protein functions from SGD[104] and classifying hits into defined groups (Supplementary Fig. 4c, d).

## Statistics and reproducibility

Results are presented either as line graphs with mean ± standard error of the mean (s.e.m.), dot plots (if $n \leq 10$), where mean (square), median (center line) and s.e.m are shown, box plots (if $n > 10$), where mean (square), median (center line) and s.e.m are shown, or as violin plots (for visualization of a high number of cells) with mean (square) and median (center line) and whiskers with minima and maxima within 1.5 interquartile range (IQR). Outliers were identified by the 1.5-IQR labeling rule. A Shapiro Wilk's test was used to test for normal distribution of data and a Levene's test for homogeneity of variances. Comparison between two groups was performed with a two-sided Student's *t*-test (Fig. 3g; Fig. 6b-Ctrl., 6b-Sui2; Supplementary Fig. 7b) and a one-way ANOVA with a Tukey post hoc (Fig. 1e; Fig. 2b, c, e; Fig. 3d; Fig. 8g; Supplementary Figs. 3c–8h; Supplementary Fig. 7h) test was applied for comparison between three or more groups. Non-normally distributed data were analyzed with either a Wilcoxon-Mann-Whitney test (Fig. 2j; Fig. 5e; Fig. 6b-Tif5; Fig. 7j) or a Kruskal Wallis test combined with a Wilcoxon signed rank test (Fig. 1f; Fig. 6d; Fig. 8c, e; Supplementary Fig. 3c-24h; Supplementary Fig. 6b). A two-way ANOVA mixed design with a Bonferroni post hoc test (Fig. 1g; Fig. 2h; Fig. 5g, h; Fig. 7b) was applied to compare different strains over time (between-subject or within-subject factors). Significances are indicated with asterisks: $**p < 0.001$, $**p < 0.01$, $*p < 0.05$, n.s. not significant in each figure panel, and detailed statistical information (degrees of freedom, statistical tests used, *p*-values of all comparisons) is given in Supplementary Table 3. For all experiments, at least three biological replicates (different yeast clones, inoculated into separate culture flasks) were used. Wherever possible, results were reproduced, and a representative experiment with

indicated number of biological replicates is shown. Microscopic experiments (Fig. 1a–d; Fig. 2a, d, g, i; Fig. 3c, e, f, h; Fig. 5d, f; Fig. 6a, c, e, g, h; Fig. 7c, h, i; Fig. 8d, f; Supplementary Fig. 1; Supplementary Fig. 2a, c; Supplementary Fig. 3b; Supplementary Fig. 5d; Supplementary Fig. 6d; Supplementary Fig. 7f, g) and biochemical assays (Supplementary Fig. 3c; Supplementary Fig. 6b) were done with indicated number of biological replicates and were repeated in three independent experiments with similar results. Refolding assays (Fig. 7d, e) were repeated in four independent experiments with similar results. Experiments involving immunoblotting (Fig. 2f; Fig. 5h; Fig. 7f; Supplementary Fig. 5c) and growth analyses (Fig. 8c; Supplementary Fig. 7c, d, e) were repeated three times with similar results. Assessments of translational activity (Fig. 8a, b; Supplementary Fig. 7a) were repeated twice with similar results. Proteomics measurements (Fig. 4c; Fig. 5b), genomic screening (Fig. 3b) and electron microscopy experiments (Fig. 1g; Fig. 7a, b; Supplementary Fig. 6a) were performed once with indicated number of biological replicates. Sample size information can be found in respective figure legends. Statistical analysis was conducted in R Studio (v.1.4.17.17), and all figures were processed with InkScape (v.1.2.2).

### Reporting summary

Further information on research design is available in the Nature Portfolio Reporting Summary linked to this article.

## Data availability

Mass spectrometry data generated in this study have been deposited in the ProteomeXchange Consortium database via the PRIDE partner repository under the accession code PXD042987 for the BioID-MS data and under PXD043093 for the CoIP-MS data as well as UV-Crosslink-MS data. Source data are provided with this paper.

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

## Acknowledgements

This work was supported by the Austrian Science Fund FWF (J4342-B21 to V.K., J4398-B to A.K.), the Swedish Research Council Vetenskapsrådet (2019-05249 to S.B., 2019-04004 to J.L.H. and 2019-04052 to C.A.), the Deutsche Forschungsgemeinschaft (213249687 and 325871075 to A.I.), the Knut and Alice Wallenberg foundation (2017.009 to T.N., J.L.H., M.O., C.A., and S.B.), Stiftelsen Olle Engkvist Byggmästare (207-0527 to S.B.), and Cancerfonden (211865 to J.L.H., 201045 to C.A., and 222488 to S.B.). pIM701 was a gift from Jiří Hašek (RRID: Addgene_74642). The authors acknowledge support from the Clinical Proteomics Mass Spectrometry at Karolinska University Hospital and Science for Life Laboratory as well as from the Protein Analytics Unit at the Biomedical Center, Ludwig-Maximilians University Munich, for providing assistance in mass spectrometry and data analysis.

## Author contributions

Conceptualization, V.K., C.A., S.B.; Methodology, V.K., A.K., C.A., S.B.; Investigation, V.K., A.K., S.G., L.L.B., X.H.; Formal Analysis, V.K., A.K., L.L.B., A.I.; Writing – Original Draft, V.K., C.A., S.B.; Writing – Review & Editing, C.A. and S.B.; Visualization, V.K., A.K.; Funding Acquisition, V.K., A.K., A.I., J.L.H., M.O., T.N., C.A., S.B.; Resources, A.I., J.L.H., M.O., T.N., C.A., S.B.; Supervision, J.L.H., M.O., T.N., C.A., S.B. All authors read and approved the final version of the manuscript.

## Funding

## Competing interests

The authors declare no competing interests.
