## [Peer Review File · Nature Communications]

Nuclear Hsp104 safeguards the dormant translation machinery during quiescenceREVIEWER COMMENTS

Reviewer #1 (Remarks to the Author):

The AAA+ disaggregase Hsp104 rescues aggregated proteins and protects cells during severe stress conditions. Here, the authors report on a novel Hsp104 function in quiescent yeast cells. They show that Hsp104 is specifically targeted to the nucleus during aging. Nuclear targeting requires the disordered C-terminal tail of Hsp104, mitochondrial respiration and reduced cytosolic translation. How these factors orchestrate nuclear transport of Hsp104 remains unclear. Interactome analysis of nuclear Hsp104 identified translation factors including eIF2 as potential substrates. This implies a role of Hsp104 in maintaining translation factors in a functional state during aging or in recovering translation factors from an aggregated state upon replenishment with nutrients. Indeed, nuclear Hsp104 allows for faster outgrowth of yeast cells and regain of translational capacity.

This is a well-executed and well-written study that reports on a novel role of Hsp104 in proteostasis. The suggested model is appealing, however, direct evidence for an Hsp104 activity towards identified translation factors is currently missing, representing a major limitation. Furthermore, *in vivo* experiments demand for analysis of controls as described below. The authors are asked to address these critical points in a revised manuscript.

Major points:

- Components of the translation machinery are described as specific substrates of nuclear Hsp104. Accordingly, they accumulate during aging in the nucleus. The author's model predicts that Hsp104 either keeps these factors in a functional state or mobilizes them during recovery phases. Both scenarios have not been tested by monitoring the fate of fluorescent translation factors (e.g. Sui2-mCherry, Fig. 5d) in non-nuclear Hsp104* or hsp104 ko cells. It is strongly recommended to determine changes in e.g. Sui2-mCherry localization, testing whether loss of nuclear Hsp104 leads to translation factor aggregation and/or the inability to retarget nuclear localized translation factors to the cytosol upon re-addition of nutrients. Aggregation of translation factors during aging or upon loss of Hsp104 should also be tested by immunoblot analysis.
- Fig. 8c: Cellular outgrowth of Hsp104 and Hsp104* cells is similar if cells were not treated with MG132. This is somewhat disappointing as Hsp104 and translation factors accumulate in the nucleus during aging in absence of MG132 treatment. The authors are asked to monitor the localization of Sui2-mCherry (or other interacting translation factors) upon MG132 treatment. Does proteasome inhibition lead to their aggregation?
- All functional assays were performed with Hsp104-GFP fusion constructs. The authors need to document that differences between Hsp104 and Hsp104* also exist for non-tagged proteins for at least in one of the performed *in vivo* experiments.

- The C-terminal tail of Hsp104 was identified as nuclear targeting signal. It is suggested to fuse this tail to GFP, testing whether it is sufficient for nuclear transport.
- Fig. 7d: The difference between nuclear and non-nuclear Hsp104 is only apparent at later timepoints of Luciferase recovery and thus considered minor. Furthermore, aged yeast cells show improved substrate reactivation in hsp104 ko cells. This surprising finding remains unexplained. It is also suggested to test whether heat shock (42°C) changes Hsp104 localization, potentially leading to nuclear enrichment of Hsp104*, explaining the minor differences in activity.

Further points:

- The C-terminal tail includes an EEVD-type motif that mediates interaction of Hsp70 and Hsp90 with TPR co-chaperones. The authors may want to discuss the possibility that such co-chaperones play roles in nuclear targeting or activity of Hsp104.
- Fig. 8b: The difference in 35S-methionine incubation between Hsp104 and Hsp104* seems much larger than results depicted in Fig. 8a. Furthermore, MG132 addition seems to increase translational capacity in Hsp104* cells. The authors should discuss this unexpected result.
- The authors suggest that during stationary phase protein aggregates predominantly form in the nucleus of yeast cells (page 12, lane 306). Evidence for this appealing model is largely missing. Nuclear aggregation should trigger foci formation of fluorescent Hsp70 chaperones. Has such change in localization been observed by the authors?
- Fig. 7e: Why did the authors analyse the amount of protein aggregates by Ponceau S staining of membranes but not Coomassie-staining of SDS-gels? The amount of protein aggregates (MG132 treatment, Hsp104*) is higher than the amount of total proteins. Does that mean the majority of cellular proteins aggregate under these conditions? This scenario is considered unlikely.
- Fig. 8c: Analysis of hsp104 ko cells is needed in these experiments as reference.
- Lane 338: change “thug-of-war” to “tug-of-war”

Reviewer #2 (Remarks to the Author):

In this manuscript, the authors use high content microscopy and quantitative proteomics to investigate a disaggregase Hsp104 during aging. The authors report that switching to respiratory metabolism and a decrease in protein biosynthesis directs cytosolic Hsp104 to the nucleus. Hsp104 interacts with eIF2 translation initiation subcomplexes and suppresses protein aggregation. Preventing Hsp104 from entering the nucleus in quiescent cells delays cell cycle re-entry because of an inability to resume protein synthesis.

The manuscript presents an interesting and novel model about the role of Hsp104 in aging. The data are mostly convincing and well-presented. Some comments/questions are listed below.

Comments

1. The authors should explain that the studies were performed yeast in the title or the abstract and earlier in the introduction.
2. In the abstract, “The nucleus has emerged as a key quality control compartment that unloads the cytosolic protein biosynthesis system from misfolded proteins.” What does “unloads” mean?
3. Fig 2: What is the role of the C terminus in determining nuclear localization? The authors identify these amino acids as being important but don’t seem to clarify what they do. Do the authors know more about this?
4. Fig 3: How many genes are in the deletion library? How many of the genes resulted in a strong, comparable and weak Hsp104 nuclear localization?
5. Fig 3: Are the STRING enrichments for specific GO terms statistically significant?
6. Fig 3: How does respiration trigger Hsp104 nuclear localization? Do the authors know any more about this?
7. Fig 4b: What is the reason that GFP nanobodies were used? Why was this advantageous over typical antibodies for Co-IP?
8. Fig 4e: It’s very hard to tell from Fig 4e which proteins were detected with which method. The authors may want to consider another way of representing this data
9. Fig 5b: Are the authors concluding that eIF2b/Sui3 does not interact with Hsp104, but eIF2a/Sui2 and eIF2g/Gcd11 do interact with Hsp104, exclusively based on the absence of these proteins from the proteomics data. It can be difficult to interpret the absence of a protein in a proteomics dataset. It would be best to directly test whether these interactions occur with a complementary method.

10. Do the translation factors such as eIF2 components leave the nucleus when quiescent cells are stimulated to divide?

11. The authors seem to be arguing that there is more Hsp104 in yeast when they are old, and that Hsp104 has the capacity to unfold proteins. But the authors also show that there are more aggregated proteins in the nucleus in older yeast. If Hsp104 is nuclearly localized in aging yeast, should the levels of aggregated proteins be lower? How do the authors think about this?

12. How did the authors validated that knockout of *doa10* and *ubr1* is resulting in an increase of misfolded proteins in the nucleus and not the cytoplasm?

13. Are the authors suggesting that Hsp104 is required to properly fold eIF2a and its binding partners themselves in the nucleus of aged cells? If so, is the problem with Hsp104-mutant cells a lack of properly folded eIF2a protein? Can the slower re-entry associated with Hsp104 mutation be reversed by overexpressing eIF2a and its partners?

Reviewer #3 (Remarks to the Author):

The authors Kohler et al present in their manuscript "Nuclear Hsp104 safeguards the dormant translation machinery during quiescence" a functional mechanism for protein folding in the nucleus by the yeast protein Hsp104. First, the authors show that Hsp104 localizes to the nucleus in aged yeast cells, due to a particular amino acid sequence on the C-terminus of the protein. Second, the authors show that relocalization of Hsp104 happens upon inhibition of protein translation, regardless of age. Third, the authors separately show that protein aggregates accumulate during aging, and that nuclear-localized Hsp104 reactivates denatured proteins in heat-shocked yeast cells, which together suggests that Hsp104 manages misfolded proteins in the nucleus. In this review, I will particularly focus on my area of expertise, which are the data and conclusions made from quantitative proteomics techniques.

-- In the localization motif studies (Fig 2c), could the authors expand on the structure of this motif? Did they attempt other mutations besides residue 901 and deletion of the 8 C-terminal residues? Are there any structure predictions for this tail region, e.g. via AlphaFold?

-- Messner et al recently performed a proteomic study of yeast gene deletions (<https://doi.org/10.1016/j.cell.2023.03.026>). Is there any existing data such as from that work which shows the effects of removing Hsp104 in support of this study's conclusions? The authors here perform a

genome-wide deletion study, but I did not see a study with deletion of Hsp104, or a particular call-out of the Hsp104 deletion results within the genome-wide deletion study. I'm curious what happens when Hsp104 is removed entirely!

-- I don't find the STRING hairballs in Fig 4c, 4d, or 4e to be particularly compelling. These concepts are difficult to make meaningful figures out of, but perhaps a volcano plot for figure 4e showing the ratio of Fc for each? For example, if both BioID and CoIP achieved a Fc of 2, then their ratio would be 1 (ideal? As it shows orthogonal validation). Perhaps for figure 4c and 4d, combining them into an UpSet plot (Venn diagram alternative) showing all shared and separate hits from each experiment, with a second UpSet plot showing the translation-annotated proteins? Just suggestions, since I don't find these hairballs to be very informative. (I do find the hairball on Fig 5b and Euler diagram on 5c to be more informative!)

-- Not a major concern, but just a note that this reviewer greatly appreciates the authors' deposition of data and code in the appropriate repositories! The submissions to ProteomeXchange appear complete, but I would request a file map (RAW file names to experimental metadata) be uploaded in addition to the excellent RAW and processed files that are currently deposited.

Point-by-point reply

We would like to thank the reviewers for the time and effort spent on our manuscript. We appreciate the constructive and valuable critiques that have been raised and have performed multiple new experiments to carefully address each comment. The new data support and strengthen our initial conclusions. Please find below the detailed point-by-point replies.

All mass spectrometry proteomics data have been deposited to the ProteomeXchange Consortium via the PRIDE partner repository.

BioID-MS data: dataset identifier PXD042987.

Reviewer account details for peer review:

Username: reviewer_pxd042987@ebi.ac.uk; Password: ncLoZnB6

CoIP-MS data as well as UV-Crosslink-MS data: dataset identifier PXD043093.

Reviewer account details for peer review:

Username: reviewer_pxd043093@ebi.ac.uk; Password: vHJP4eKg

Reviewer #1 (Remarks to the Author):

The AAA+ disaggregase Hsp104 rescues aggregated proteins and protects cells during severe stress conditions. Here, the authors report on a novel Hsp104 function in quiescent yeast cells. They show that Hsp104 is specifically targeted to the nucleus during aging. Nuclear targeting requires the disordered C-terminal tail of Hsp104, mitochondrial respiration and reduced cytosolic translation. How these factors orchestrate nuclear transport of Hsp104 remains unclear. Interactome analysis of nuclear Hsp104 identified translation factors including eIF2 as potential substrates. This implies a role of Hsp104 in maintaining translation factors in a functional state during aging or in recovering translation factors from an aggregated state upon replenishment with nutrients. Indeed, nuclear Hsp104 allows for faster outgrowth of yeast cells and regain of translational capacity.

This is a well-executed and well-written study that reports on a novel role of Hsp104 in proteostasis. The suggested model is appealing, however, direct evidence for an Hsp104 activity towards identified translation factors is currently missing, representing a major limitation. Furthermore, in vivo experiments demand for analysis of controls as described below. The authors are asked to address these critical points in a revised manuscript.

Major points:

Components of the translation machinery are described as specific substrates of nuclear Hsp104. Accordingly, they accumulate during aging in the nucleus. The author's model predict that Hsp104 either keeps these factors in a functional state or mobilizes them during recovery phases. Both scenarios have not been tested by monitoring the fate of fluorescent translation factors (e.g. Sui2-mCherry, Fig. 5d) in non-nuclear Hsp104* or hsp104 ko cells. It is strongly recommended to determine changes in e.g. Sui2-mCherry localization, testing whether loss of nuclear Hsp104 leads to translation factor aggregation and/or the inability to retarget nuclear localized translation factors to the cytosol upon re-addition of nutrients. Aggregation of translation factors during aging or upon loss of Hsp104 should also be tested by immunoblot analysis.

We have performed the suggested experiments and now show that nuclear Hsp104 controls the nucleo-cytoplasmic partitioning of eIF2 α /Sui2 in response to aging as well as upon cell cycle re-entry upon refeeding. Specifically, microscopic analysis of cells carrying GFP-tagged Hsp104 variants and Sui2-mCherry revealed that nuclear targeting of Sui2 in aged cells is facilitated by the presence of nuclear Hsp104. Respective quantification of nuclear accumulation of Sui2 using either intensity ratios between the nucleus and cytosol (Figure 5d, e and Supplementary Figure 3d for visualization) or percentage of cells with Sui2 nuclear accumulation (Figure 5f, g) have now been included. Corresponding text in the Results on page 9 has been modified accordingly:

“This revealed that eIF2 α /Sui2 was evenly distributed in young, dividing cells but progressively accumulated in the nucleus as cells aged, resembling the subcellular distribution of Hsp104 (Fig. 5d, e and Supplementary Fig 5d). Interestingly, we observed significantly reduced nuclear targeting of Sui2 in aged Hsp104 cells, assessed via determination of the nuclear/cytoplasmic intensity ratios as well as by quantification of the fraction of cells with nuclear accumulation of Sui2 (Fig. 5d-g and Supplementary Fig 5d). Sui2 protein levels were comparable in cells expressing either Hsp104 or Hsp104* during aging, ruling out that the changed localization is a consequence of altered expression levels (Fig. 5h). In sum, nuclear Hsp104 interacts with eIF2 α /Sui2 and regulates its nucleo-cytoplasmic partitioning in aging cells”.*

Direct testing of the nucleo-cytoplasmic partitioning of Sui2 upon addition of fresh media to aged cells demonstrated that Sui2 rapidly exits the nucleus dependent on nuclear Hsp104. Micrographs as well as corresponding quantifications of this new key finding have been included in Figure 8d, e. Moreover, we show that targeting of Sui2 to the nucleus protects it from stress-induced aggregation (Figure 8f, g). The following text has been added to the Results on page 14:

“Importantly, replenishment of nutrients and thus cell cycle re-entry triggered the rapid nuclear exit of Sui2 only in the presence of nuclear Hsp104 (Fig. 8d, e). We reasoned that targeting of Sui2 to the nucleus protects it from aggregation. Indeed, perturbing proteostasis by long-term proteasomal impairment (Bortezomib treatment) caused Sui2 to aggregate particularly in cells lacking nuclear Hsp104 (Fig. 8f, g). Upon nutrient replenishment, cells were able to disentangle these aggregates (Fig. 8f, g). Thus, nuclear Hsp104 facilitates efficient nuclear targeting of Sui2 when translational activity decreases in aged cells, protects it from stress-induced aggregation and finally supports nuclear exit of Sui2 upon resumption of translation and growth.”

Fig. 8c: Cellular outgrowth of Hsp104 and Hsp104* cells is similar if cells were not treated with MG132. This is somewhat disappointing as Hsp104 and translation factors accumulate in the nucleus during aging in absence of MG132 treatment. The authors are asked to monitor the localization of Sui2-mCherry (or other interacting translation factors) upon MG132 treatment. Does proteasome inhibition lead to their aggregation?

As described above, we have now performed additional experiments that show that nuclear Hsp104 controls the nucleo-cytoplasmic partitioning of eIF2 α /Sui2 in response to aging. We in addition find that the aggregation status of Sui2 is dependent on its nuclear targeting when the proteasome is impaired (Figure 8f, g). The following text has been added to the Results on page 14:

“We reasoned that targeting of Sui2 to the nucleus protects it from aggregation. Indeed, perturbing proteostasis by long-term proteasomal impairment (Bortezomib treatment) caused Sui2 to aggregate particularly in cells lacking nuclear Hsp104 (Fig. 8f, g). Upon nutrient replenishment, cells were able to disentangle these aggregates (Fig. 8f, g). Thus, nuclear Hsp104 facilitates efficient nuclear targeting of Sui2 when translational activity decreases in aged cells, protects it from stress-induced aggregation and finally supports nuclear exit of Sui2 upon resumption of translation and growth.”

In this regard, we added the following text to the Discussion on page 16:

“When translation rates drop during aging, nuclear Hsp104 facilitates the targeting of eIF2 α /Sui2 to the nucleus and protects it from stress-induced aggregation. Upon nutrient replenishment and thus resumption of translation, Hsp104 promotes nuclear exit of Sui2.”

All functional assays were performed with Hsp104-GFP fusion constructs. The authors need to document that differences between Hsp104 and Hsp104* also exist for non-tagged proteins for at least in one of the performed in vivo experiments.

We have now added two additional experiments based on non-tagged Hsp104 and Hsp104* strains. In Supplementary Figure 7a, b, we assessed translational activity at day 1, day 2 and day 3 and in Supplementary Figure 7c, the regrowth of cells aged to day 5 upon replenishment with fresh nutrients was followed. The experiments substantiate our findings. We have included the following text in the Results on page 13:

“Interestingly, nuclear but not cytosolic Hsp104 was required for the rapid resumption of translation after an extended period of aging (Fig. 8a). Similar results were obtained using strains with untagged Hsp104 and Hsp104 variants (Supplementary Fig. 7a, b).”*

“MG-132 pre-treatment prominently delayed resumption of growth, in particular in the absence of nuclear Hsp104 (Fig. 8c and Supplementary Fig. 7c).”

The C-terminal tail of Hsp104 was identified as nuclear targeting signal. It is suggested to fuse this tail to GFP, testing whether it is sufficient for nuclear transport.

We performed the suggested experiment using the last 8 amino acids (AA901-908) of the Hsp104 and Hsp104* C-terminal tail. The fusion of the wild type Hsp104 tail (but not the Hsp104* mutant) targeted GFP to the nucleus selectively in aged cells, suggesting that this minimal motif is sufficient to allow age-dependent relocalization to the nucleus. Respective micrographs and accompanying quantifications have been included in Figure 2i, j, and the following text has been added to the Results on page 6/7:

“Notably, fusing the last 8 amino acids of Hsp104, corresponding to the TPR protein binding motif, to GFP was sufficient to target the chimera to the nucleus in an age-dependent manner

(Fig. 2j, i). The M901V mutation abrogated nuclear accumulation. Hence, the short C-terminal motif is required and sufficient for the age-induced targeting of Hsp104 to the nucleus.”

Fig. 7d: The difference between nuclear and non-nuclear Hsp104 is only apparent at later timepoints of Luciferase recovery and thus considered minor. Furthermore, aged yeast cells show improved substrate reactivation in hsp104 ko cells. This surprising finding remains unexplained. It is also suggested to test whether heat shock (42°C) changes Hsp104 localization, potentially leading to nuclear enrichment of Hsp104*, explaining the minor differences in activity.

We have performed the suggested heat shock experiment and although we observe extensive aggregation, this does not seem to impact on Hsp104 nuclear versus cytosolic localisation (now included in Supplementary Figure 6c). We propose the involvement of alternative disaggregation pathways, for example the recently described Hsp70-Apj1 system that is active in the nucleus of yeast cells. Additionally, we tested the basal activity of the heat shock response transcription factor Hsf1 and found no difference between cells expressing Hsp104 and Hsp104* (now included in Supplementary Figure 6b). We included the following text into the respective Results section on page 11 and page 12:

“To assess a potential cytoprotective function of nuclear Hsp104, we combined a perturbation of the proteostasis system with the exclusion of Hsp104 from the nucleus. Initially, we established that the age-dependent accumulation of Hsp104 in the nucleus did not impact on the basal transcriptional activity of the heat shock response transcription factor Hsf1 (Supplementary Fig. 6b). We perturbed the proteostasis system by heat shocking cells with nuclear and nuclear-targeting defective Hsp104. Heat shock induced numerous aggregates decorated by Hsp104 or Hsp104* that over time were disentangled, yet did not affect the overall nuclear versus cytoplasmic localization of the respective variant (Supplementary Fig. 6c).”*

“We also observed substantial Hsp104-independent reactivation of ^{GFP}FFL-NLS particularly in aged cells, suggesting that other disaggregation systems were activated, such as nuclear Hsp70 that works in concert with the J-domain protein Apj1 ⁶⁵.”

Further points:

The C-terminal tail includes an EEVD-type motif that mediates interaction of Hsp70 and Hsp90 with TPR co-chaperones. The authors may want to discuss the possibility that such co-chaperones play roles in nuclear targeting or activity of Hsp104.

We have now added the following sentences in the Results and the Discussion:

Results on page 6:

“Sequencing revealed a single methionine to valine substitution (M901V) localized to the C-terminal tail of Hsp104, a suggested binding site for tetratricopeptide repeat (TPR) proteins ^{50,51}, in the strain from the Yeast GFP clone collection (Fig. 2c).”

Results on page 6/7:

“Notably, fusing the last 8 amino acids of Hsp104, corresponding to the TPR protein binding motif, to GFP was sufficient to target the chimera to the nucleus in an age-dependent manner (Fig. 2j, i). The M901V mutation abrogated nuclear accumulation. Hence, the short C-terminal motif is required and sufficient for the age-induced targeting of Hsp104 to the nucleus.”

Discussion on page 16/17 has been revised:

“These interactions were absent in the M901V mutant (Hsp104), highlighting a critical role for the C-terminal TPR protein binding motif to engage the nuclear targeting machinery. Indeed, this peptide is sufficient to target GFP to the nucleus of specifically aged cells. Thus, we find that the C-terminus of Hsp104 has two distinct functional interactions: (i) association with the nuclear targeting machinery and (ii) direct binding to its eIF2 complex clients. Indeed, many chaperones, including Hsp70, Hsp90 and J-domain proteins, employ their TPR protein binding motifs located to their extreme C-termini for various and complex interactions within the proteostasis network⁷⁶⁻⁸⁴. Our finding that a minimal TPR protein binding motif is required and sufficient for age-dependent nuclear targeting of Hsp104 suggests that this process is controlled by interactions with one or several TPR proteins. Indeed, TPR proteins have previously been associated with nucleo-cytoplasmic shuttling^{51,85-89}. The hexameric conformation of Hsp104 allows for multiple associations with the six flexible C-termini. Nuclear targeting of Hsp104 does not depend on Hsp70 or its own disaggregation activity, ruling out additional piggy-backing on nuclear-targeted misfolded proteins. Yet, Hsp104 localization appears to be partially regulated by a tug-of-war between its various clients and TPR protein interactions. For instance, blocking mitochondrial respiration and hence mitochondrial protein import, which triggers cytosolic aggregation of mitochondrial precursor proteins^{90,91}, is sufficient to extract Hsp104 from the nucleus of aged cells.”*

Fig. 8b: The difference in 35S-methionine incubation between Hsp104 and Hsp104* seems much larger than results depicted in Fig. 8a. Furthermore, MG132 addition seems to increase translational capacity in Hsp104* cells. The authors should discuss this unexpected result.

We quantified the difference between Hsp104 and Hsp104* in both experiments. We found that the radiolabelled signals in Hsp104* cells correspond to 70% and 52% of Hsp104 cells, respectively (comparing Figure 8a and 8b). Thus, both experiments show a clear drop in protein synthesis upon Hsp104 mutation at a comparable level.

When quantifying protein synthesis, we found that MG-132 treated Hsp104* cells showed a slightly decreased translational activity compared to untreated variants (0.47 vs. 0.52). We added the respective values to the autoradiogram (Figure 8b).

The authors suggest that during stationary phase protein aggregates predominantly form in the nucleus of yeast cells (page 12, lane 306). Evidence for this appealing model is largely missing. Nuclear aggregation should trigger foci formation of fluorescent Hsp70 chaperones. Has such change in localization been observed by the authors?

We have now clarified our text section on the emergence of nuclear aggregates with cellular age. We have rephrased our text to make clear that we do not suggest that aggregates exclusively or

predominantly form in the nucleus during aging. Instead, we state that cellular aging drives the emergence of aggregates also in the nucleus, which is absent in young cells. In addition to the quantification of nuclear aggregates detected by EM (included in the original manuscript), we now have added additional fluorescence microscopy data showing that Hsp104 forms foci in the nucleus of aged cells, indicative of the emergence of aggregates (now included in Figure 7c). The revised text on page 11 now reads:

“First, we asked whether aggregates accumulate in the nucleus during cellular aging. Indeed, transmission electron microscopy showed that electron-dense clusters (EDCs), likely reflecting protein aggregates, appeared in the nuclei of aged but not that of young cells, with 61% of the aged population scoring EDC-positive (Fig. 7a). Importantly, immunogold labeling demonstrated a clear co-localization of native Hsp104 with these clusters (Fig. 7b, Supplementary Fig. 6a for unlabeled micrographs). Likewise, fluorescence microscopy demonstrated the accumulation of nuclear aggregates decorated by Hsp104-GFP upon prolonged aging (Fig. 7c). Hence, cellular aging is accompanied by the emergence of protein aggregates also in the nucleus.”

Fig. 7e: Why did the authors analyse the amount of protein aggregates by Ponceau S staining of membranes but not Coomassie-staining of SDS-gels? The amount of protein aggregates (MG132 treatment, Hsp104*) is higher than the amount of total proteins. Does that mean the majority of cellular proteins aggregate under these conditions? This scenario is considered unlikely.

We have now updated the respective Figure legend (former Figure 7e, now Figure 7f) to clarify that we have loaded an increased corresponding amount (18x more) of the aggregate fraction compared to the total lysate fraction to ensure detectable signals. Hence, the aggregated proteins only reflect a subfraction of the total protein. Both methods, Ponceau S staining of membranes and Coomassie staining of SDS gels are suitable for total protein detection. As we did not have any preferences, we decided to detect via Ponceau S on the membrane.

Fig. 8c: Analysis of hsp104 ko cells is needed in these experiments as reference.

We repeated this experiment and added the *hsp104Δ* strain as a control (Figure 8c). The results support our original conclusions. The *hsp104Δ* strain grows as the Hsp104* expressing strain, supporting that indeed Hsp104 is required for timely and efficient re-entry of aged cells into the cell cycle. The following sentence has been added to the Results on page 13/14:

*“We followed regrowth after supplementation of fresh nutrients. MG-132 pre-treatment prominently delayed resumption of growth, in particular in the absence of nuclear Hsp104 (Fig. 8c and Supplementary Fig. 7c). Interestingly, cells lacking Hsp104 (*hsp104Δ*) displayed a similar growth delay as Hsp104* expressing cells (Fig. 8c). While wild type Hsp104 cells completed the first cell cycle within 9.4 h, *hsp104Δ* cells required 10.3 h for the first doubling and Hsp104* cells needed 10.9 h, suggesting that Hsp104* causes a loss-of-function phenotype in respect to cell cycle re-entry. The subsequent exponential growth rates did not differ between the strains.”*

- Lane 338: change “thug-of-war” to “tug-of-war”

We apologize, this error was corrected.

Reviewer #2 (Remarks to the Author):

In this manuscript, the authors use high-content microscopy and quantitative proteomics to investigate a disaggregase Hsp104 during aging. The authors report that switching to respiratory metabolism and a decrease in protein biosynthesis directs cytosolic Hsp104 to the nucleus. Hsp104 interacts with eIF2 translation initiation subcomplexes and suppresses protein aggregation. Preventing Hsp104 from entering the nucleus in quiescent cells delays cell cycle re-entry because of an inability to resume protein synthesis.

The manuscript presents an interesting and novel model about the role of Hsp104 in aging. The data are mostly convincing and well-presented. Some comments/questions are listed below.

Comments

The authors should explain that the studies were performed yeast in the title or the abstract and earlier in the introduction.

We agree and now mention “yeast” in the abstract twice.

In the abstract, “The nucleus has emerged as a key quality control compartment that unloads the cytosolic protein biosynthesis system from misfolded proteins.” What does “unloads” mean?

We have exchanged this sentence in the abstract. It now reads:

“The nucleus has emerged as a key quality control compartment that handles misfolded proteins produced by the cytosolic protein biosynthesis system.”

Fig 2: What is the role of the C terminus in determining nuclear localization? The authors identify these amino acids as being important but don’t seem to clarify what they do. Do the authors know more about this?

This comment is overlapping with questions brought up from all three reviewers. As outlined above, we have made the following new experiments and text changes in this respect:

We performed additional experiments using the last 8 amino acids (AA901-908) of the Hsp104 and Hsp104* C-terminal tail. The fusion of the wild type Hsp104 tail (but not the Hsp104* mutant) targeted GFP to the nucleus selectively in aged cells, suggesting that this minimal motif is sufficient to allow age-dependent relocalization to the nucleus. Respective micrographs and accompanying quantifications have been included in Figure 2i, j, and the following text has been added to the Results on page 6/7:

“Notably, fusing the last 8 amino acids of Hsp104, corresponding to the TPR protein binding motif, to GFP was sufficient to target the chimera to the nucleus in an age-dependent manner (Fig. 2j, i). The M901V mutation abrogated nuclear accumulation. Hence, the short C-terminal motif is required and sufficient for the age-induced targeting of Hsp104 to the nucleus.”

In addition, we have added the following text in respect to TPR binding proteins in the Results and the Discussion:

Results on page 6:

“Sequencing revealed a single methionine to valine substitution (M901V) localized to the C-terminal tail of Hsp104, a suggested binding site for tetratricopeptide repeat (TPR) proteins^{50,51}, in the strain from the Yeast GFP clone collection (Fig. 2c).”

Discussion on page 16/17 has been revised:

“These interactions were absent in the M901V mutant (Hsp104), highlighting a critical role for the C-terminal TPR protein binding motif to engage the nuclear targeting machinery. Indeed, this peptide is sufficient to target GFP to the nucleus of specifically aged cells. Thus, we find that the C-terminus of Hsp104 has two distinct functional interactions: (i) association with the nuclear targeting machinery and (ii) direct binding to its eIF2 complex clients. Indeed, many chaperones, including Hsp70, Hsp90 and J-domain proteins, employ their TPR protein binding motifs located to their extreme C-termini for various and complex interactions within the proteostasis network^{76–84}. Our finding that a minimal TPR protein binding motif is required and sufficient for age-dependent nuclear targeting of Hsp104 suggests that this process is controlled by interactions with one or several TPR proteins. Indeed, TPR proteins have previously been associated with nucleo-cytoplasmic shuttling^{51,85–89}. The hexameric conformation of Hsp104 allows for multiple associations with the six flexible C-termini. Nuclear targeting of Hsp104 does not depend on Hsp70 or its own disaggregation activity, ruling out additional piggy-backing on nuclear-targeted misfolded proteins. Yet, Hsp104 localization appears to be partially regulated by a tug-of-war between its various clients and TPR protein interactions. For instance, blocking mitochondrial respiration and hence mitochondrial protein import, which triggers cytosolic aggregation of mitochondrial precursor proteins^{90,91}, is sufficient to extract Hsp104 from the nucleus of aged cells.”*

Fig 3: How many genes are in the deletion library? How many of the genes resulted in a strong, comparable and weak Hsp104 nuclear localization?

We have now included this information in the respective figure (Figure 3a). There are approximately 4800 mutants in the yeast deletion collection. Out of these mutants, 173 displayed a strongly increased nuclear accumulation of Hsp104 and 230 resulted in weak nuclear accumulation.

Fig 3: Are the STRING enrichments for specific GO terms statistically significant?

Yes, the enrichment clusters are significant, we added this information to the respective material and methods chapter. The respective section on page 29 now reads as follows:

“Hits enriched in Hsp104 eluates from BioID, CoIP and UV-crosslinking experiments were subjected to network analyses using Cytoscape (v.3.9.1), either using fold change values for visualization or subjecting genes to STRING analyses and Markov algorithm calculations to identify statistically significant enrichment clusters⁵².”

Fig 3: How does respiration trigger Hsp104 nuclear localization? Do the authors know any more about this?

As switch to respiration is associated with decreased translation. We have performed additional experiments that lend support to the notion that it is translation rather than respiration that is the most downstream cue for nucleo-cytoplasmic shuttling of Hsp104. Specifically, we now find that glucose addition to aged cells triggers the exit of Hsp104 from the nucleus within 1 h. This nuclear exit was dependent on the initiation of translation. This new data set has been included in Figure 6c, d and the corresponding text in the Results on page 10 has been adapted:

“Refeeding starved cells with glucose, which induces rapid translation, resulted in the exit of a significant fraction of Hsp104 from the nucleus within 1 hour (Fig. 6c, d). To test whether this depends on the restart of translation, we aged cells with GAL-promoter-regulated expression of Sui2 and Tif5 in media containing galactose, allowing for initial expression of Sui2 and Tif5 during exponential growth until galactose was depleted. In aged cells depleted of Sui2 or Tif5, the addition of glucose did not trigger the nuclear exit of Hsp104 (Fig. 6c, d), suggesting that translational restart serves as a signal for Hsp104 mobilization from the nucleus.”

Consequently, we have clarified our understanding of how respiration impacts on Hsp104 nuclear localisation by editing the Discussion on page 15:

“As nutrients become limiting, cells switch to respiratory metabolism to prepare for stationary phase and translation rates decrease^{58,59,69}, reducing the demand for Hsp104 in the cytosol. Concurrently, cells re-direct Hsp104 to the nucleus, where it associates with and disentangles protein aggregates. Hence, the decreased protein synthesis is likely the most downstream event that drives nuclear targeting of Hsp104.”

Moreover, we have directly tested the involvement of the key stress transcription factors that are activated during respiratory growth and entry into stationary phase, Msn2/4. The nuclear targeting of Hsp104 is independent of the function of these transcription factors, again pointing to translation being the most downstream regulatory event. The new data has been included in Supplementary Fig. 3b, c. The respective Results section on page 7/8 reads as follows:

“The key stress transcription factors activated during respiratory growth and entry into stationary phase, Msn2/4³⁶⁻³⁹, were not required to target Hsp104 to the nucleus (Supplementary Fig. 3b). Loss of nuclear Hsp104 did not affect Msn2/4 activity (Supplementary Fig. 3c). Collectively, the genome-wide screen revealed the presence of a Hsp104 nuclear targeting machinery and identified mitochondrial respiration, but not its associated stress response signaling, as a key metabolic cue that controls nuclear relocalization.”

Fig 4b: What is the reason that GFP nanobodies were used? Why was this advantageous over typical antibodies for Co-IP?

The recombinant and therefore highly purified GFP nanobodies from Chromotek have been advantageous for several of our projects. We find them to give less background than standard antibodies

during CoIP, they do not give signals with secondary antibodies on Western Blots and work ideally with mass spectrometry-coupled analyses.

Fig 4e: It's very hard to tell from Fig 4e which proteins were detected with which method. The authors may want to consider another way of representing this data

We thank the reviewer for this suggestion. We have now completely restructured the respective figure panel to provide more clarity (old Figure 4e, now Figure 4c, d).

Fig 5b: Are the authors concluding that eIF2b/Sui3 does not interact with Hsp104, but eIF2a/Sui2 and eIF2g/Gcd11 do interact with Hsp104, exclusively based on the absence of these proteins from the proteomics data. It can be difficult to interpret the absence of a protein in a proteomics dataset. It would be best to directly test whether these interactions occur with a complementary method.

We have performed the suggested experiments. Using high affinity ALFA-tag nanobodies, we show that not only Sui2 and Tif5 but also Sui3 interact with Hsp104. We have included this data into Supplementary Fig. 5c and changed the text on page 9 accordingly:

“Though eIF2 α /Sui3 was not present in our analyses, we detected an interaction with Hsp104 when performing pulldown experiments coupled with immunoblot analyses (Supplementary Fig. 5c).”

Do the translation factors such as eIF2 components leave the nucleus when quiescent cells are stimulated to divide?

We have performed additional experiments (also in response to a comment from reviewer #1) and now show that nuclear Hsp104 controls the nucleo-cytoplasmic partitioning of eIF2 α /Sui2 in response to aging as well as upon cell cycle re-entry upon refeeding. Specifically, microscopic analysis of cells carrying GFP-tagged Hsp104 variants and Sui2-mCherry revealed that nuclear targeting of Sui2 in aged cells is facilitated by the presence of nuclear Hsp104. Respective quantification of nuclear accumulation of Sui2 using either intensity ratios between the nucleus and cytosol (Figure 5d, e and Supplementary Figure 3d for visualization) or percentage of cells with Sui2 nuclear accumulation (Figure 5f, g) have now been included. Corresponding text in the Results on page 9 has been modified accordingly:

“This revealed that eIF2 α /Sui2 was evenly distributed in young, dividing cells but progressively accumulated in the nucleus as cells aged, resembling the subcellular distribution of Hsp104 (Fig. 5d, e and Supplementary Fig 5d). Interestingly, we observed significantly reduced nuclear targeting of Sui2 in aged Hsp104 cells, assessed via determination of the nuclear/cytoplasmic intensity ratios as well as by quantification of the fraction of cells with nuclear accumulation of Sui2 (Fig. 5d-g and Supplementary Fig 5d). [...] In sum, nuclear Hsp104 interacts with eIF2 α /Sui2 and regulates its nucleo-cytoplasmic partitioning in aging cells.”*

Direct testing of the nucleo-cytoplasmic partitioning of Sui2 upon addition of fresh media to aged cells demonstrated that Sui2 rapidly exits the nucleus dependent on nuclear Hsp104. Micrographs as well as corresponding quantifications of this new key finding have been included in Figure 8d, e. The following text has been added to the Results on page 14:

“Importantly, replenishment of nutrients and thus cell cycle re-entry triggered the rapid nuclear exit of Sui2 only in the presence of nuclear Hsp104 (Fig. 8d, e). [...] Thus, nuclear Hsp104 facilitates efficient nuclear targeting of Sui2 when translational activity decreases in aged cells, protects it from stress-induced aggregation and finally supports nuclear exit of Sui2 upon resumption of translation and growth.”

In this regard, we added the following text to the Discussion on page 16:

“When translation rates drop during aging, nuclear Hsp104 facilitates the targeting of eIF2 α /Sui2 to the nucleus and protects it from stress-induced aggregation. Upon nutrient replenishment and thus resumption of translation, Hsp104 promotes nuclear exit of Sui2.”

The authors seem to be arguing that there is more Hsp104 in yeast when they are old, and that Hsp104 has the capacity to unfold proteins. But the authors also show that there are more aggregated proteins in the nucleus in older yeast. If Hsp104 is nuclearly localized in aging yeast, should the levels of aggregated proteins be lower? How do the authors think about this?

We thank the reviewer for pointing out this apparent paradox. We have changed the discussion to clarify why we think that the rates of aggregation and disaggregation result in the accumulation of protein aggregates in aged wild type cells. The following modification has been included on page 15:

“Thus, the proteostatic damage that builds up in quiescence when nuclear Hsp104 is missing becomes evident only upon re-start of protein biosynthesis, resulting in inefficient resumption of translation as well as delayed cell cycle re-entry. Furthermore, additional proteasomal inhibition enforces this defect, suggesting that nuclear Hsp104 and the proteasome work in parallel to remove the misfolding protein damage. We find that also wild type cells accumulate detectable levels of nuclear protein aggregates during aging, likely due to the fact that the rates of Hsp104-dependent disaggregation are lower than the rates of protein aggregation.”

How did the authors validated that knockout of *doa10* and *ubr1* is resulting in an increase of misfolded proteins in the nucleus and not the cytoplasm?

We recognize that our interpretation was presented so it could be mistaken as a claim and now have rephrased the relevant sentence in the results on page 12 to clarify that misfolded protein species in cells lacking Doa10 and Ubr1 likely increase not only in the nucleus but also in the cytoplasm.

*“Likewise, genetic inactivation of key PQC E3 ubiquitin ligases (*doa10* Δ *ubr1* Δ), which remove nuclear and cytosolic misfolded proteins, increased nuclear Hsp104 accumulation in aged cells, suggesting that more misfolded protein species are present in the nucleus in this mutant (Fig. 7i, j).”*

Are the authors suggesting that Hsp104 is required to properly fold eIF2 α and its binding partners themselves in the nucleus of aged cells? If so, is the problem with Hsp104-mutant cells a lack of

properly folded eIF2a protein? Can the slower re-entry associated with Hsp104 mutation be reversed by overexpressing eIF2a and its partners?

As described above, we have now performed additional experiments that show that nuclear Hsp104 controls the nucleo-cytoplasmic partitioning of eIF2 α /Sui2 in response to aging. We in addition find that the aggregation status of Sui2 is dependent on its nuclear targeting when the proteasome is impaired (Figure 8f, g). The following text has been added to the Results on page 14:

“Importantly, replenishment of nutrients and thus cell cycle re-entry triggered the rapid nuclear exit of Sui2 only in the presence of nuclear Hsp104 (Fig. 8d, e). We reasoned that targeting of Sui2 to the nucleus protects it from aggregation. Indeed, perturbing proteostasis by long-term proteasomal impairment (Bortezomib treatment) caused Sui2 to aggregate particularly in cells lacking nuclear Hsp104 (Fig. 8f, g). Upon nutrient replenishment, cells were able to disentangle these aggregates (Fig. 8f, g). Thus, nuclear Hsp104 facilitates efficient nuclear targeting of Sui2 when translational activity decreases in aged cells, protects it from stress-induced aggregation and finally supports nuclear exit of Sui2 upon resumption of translation and growth.”

Moreover, we tested whether overexpression of Sui2 would be sufficient to restore re-growth of quiescent cells lacking nuclear Hsp104. However, the growth defect was not simply due to limiting levels of Sui2. We formally cannot exclude the possibility that co-overexpression of other eIF2 components might alleviate the phenotype. We added the new data into Supplementary Fig. 7d, e and included the following text into the Results on page 14:

“We ruled out that the delayed resumption of growth was merely the consequence of limiting cytosolic Sui2 levels by conditionally overexpressing Sui2 (Supplementary Fig. 7d, e).”

Reviewer #3 (Remarks to the Author):

The authors Kohler et al present in their manuscript “Nuclear Hsp104 safeguards the dormant translation machinery during quiescence” a functional mechanism for protein folding in the nucleus by the yeast protein Hsp104. First, the authors show that Hsp104 localizes to the nucleus in aged yeast cells, due to a particular amino acid sequence on the C-terminus of the protein. Second, the authors show that relocalization of Hsp104 happens upon inhibition of protein translation, regardless of age. Third, the authors separately show that protein aggregates accumulate during aging, and that nuclear-localized Hsp104 reactivates denatured proteins in heat-shocked yeast cells, which together suggests that Hsp104 manages misfolded proteins in the nucleus. In this review, I will particularly focus on my area of expertise, which are the data and conclusions made from quantitative proteomics techniques.

In the localization motif studies (Fig 2c), could the authors expand on the structure of this motif? Did they attempt other mutations besides residue 901 and deletion of the 8 C-terminal residues? Are there any structure predictions for this tail region, e.g. via AlphaFold?

This comment is overlapping with a question brought up by all three reviewers. As outlined above, we have addressed this with new experiments and text changes. According to current understanding (e.g., PMID: 23650362, PMID: 27478928 as well as secondary structure prediction algorithms), the C-

terminal domain of Hsp104 is unstructured and linked to co-chaperone recruitment. As outlined below, we now expand on this topic in the discussion.

We did not perform further mutagenesis than the M901V as well as complete C-terminal truncation (8 aa), but we have now included additional experiments using the last 8 amino acids (AA901-908) of the Hsp104 and Hsp104* C-terminal tail. The fusion of the wild type Hsp104 tail (but not the Hsp104* mutant) targeted GFP to the nucleus selectively in aged cells, suggesting that this minimal motif is sufficient to allow age-dependent relocalization to the nucleus. Respective micrographs and accompanying quantifications have been included in Figure 2i, j, and the following text has been added to the Results on page 6/7:

“Notably, fusing the last 8 amino acids of Hsp104, corresponding to the TPR protein binding motif, to GFP was sufficient to target the chimera to the nucleus in an age-dependent manner (Fig. 2j, i). The M901V mutation abrogated nuclear accumulation. Hence, the short C-terminal motif is required and sufficient for the age-induced targeting of Hsp104 to the nucleus.”

In addition, we have added the following text in respect to TPR binding proteins in the Results section and the Discussion:

Results on page 6:

“Sequencing revealed a single methionine to valine substitution (M901V) localized to the C-terminal tail of Hsp104, a suggested binding site for TPR (tetratricopeptide repeat) proteins^{50,51}, in the strain from the Yeast GFP clone collection (Fig. 2c).”

Discussion on page 17/18 has been revised:

“These interactions were absent in the M901V mutant (Hsp104), highlighting a critical role for the C-terminal TPR protein binding motif to engage the nuclear targeting machinery. Indeed, this peptide is sufficient to target GFP to the nucleus of specifically aged cells. Thus, we find that the C-terminus of Hsp104 has two distinct functional interactions: (i) association with the nuclear targeting machinery and (ii) direct binding to its eIF2 complex clients. Indeed, many chaperones, including Hsp70, Hsp90 and J-domain proteins, employ their TPR protein binding motifs located to their extreme C-termini for various and complex interactions within the proteostasis network^{76–84}. Our finding that a minimal TPR protein binding motif is required and sufficient for age-dependent nuclear targeting of Hsp104 suggests that this process is controlled by interactions with one or several TPR proteins. Indeed, TPR proteins have previously been associated with nucleo-cytoplasmic shuttling^{51,85–89}. The hexameric conformation of Hsp104 allows for multiple associations with the six flexible C-termini. Nuclear targeting of Hsp104 does not depend on Hsp70 or its own disaggregation activity, ruling out additional piggy-backing on nuclear-targeted misfolded proteins. Yet, Hsp104 localization appears to be partially regulated by a tug-of-war between its various clients and TPR protein interactions. For instance, blocking mitochondrial respiration and hence mitochondrial protein import, which triggers cytosolic aggregation of mitochondrial precursor proteins^{90,91}, is sufficient to extract Hsp104 from the nucleus of aged cells.”*

Messner et al recently performed a proteomic study of yeast gene deletions (<https://doi.org/10.1016/j.cell.2023.03.026>). Is there any existing data such as from that work which shows the effects of removing Hsp104 in support of this study's conclusions? The authors here perform a genome-wide deletion study, but I did not see a study with deletion of Hsp104, or a particular call-out of the Hsp104 deletion results within the genome-wide deletion study. I'm curious what happens when Hsp104 is removed entirely!

While the mentioned proteomic study is highly relevant for functional studies, all experiments were performed in young, actively dividing cells (after 8 h) in contrast to the aging phenotype that we study here. Of note, in actively dividing cells, Hsp104 does not accumulate in the nucleus, thus we cannot perform direct comparisons with the other studies.

We have revisited the experiments that unveiled our key phenotype - the delayed cell cycle re-entry of Hsp104* expressing cells upon refeeding - using an *hsp104Δ* strain (Fig. 8c). The results support our original conclusions. The *hsp104Δ* strain grows as the Hsp104* expressing strain, supporting that indeed Hsp104 is required for timely and efficient re-entry of aged cells into the cell cycle. The following sentence has been added to the Results on page 13/14:

“We followed regrowth after supplementation of fresh nutrients. MG-132 pre-treatment prominently delayed resumption of growth, in particular in the absence of nuclear Hsp104 (Fig. 8c and Supplementary Fig. 7c). Interestingly, cells lacking Hsp104 (hsp104Δ) displayed a similar growth delay as Hsp104 expressing cells (Fig. 8c). While wild type Hsp104 cells completed the first cell cycle within 9.4 h, hsp104Δ cells required 10.3 h for the first doubling and Hsp104* cells needed 10.9 h, suggesting that Hsp104* causes a loss-of-function phenotype in respect to cell cycle re-entry. The subsequent exponential growth rates did not differ between the strains.”*

I don't find the STRING hairballs in Fig 4c, 4d, or 4e to be particularly compelling. These concepts are difficult to make meaningful figures out of, but perhaps a volcano plot for figure 4e showing the ratio of Fc for each? For example, if both BioID and CoIP achieved a Fc of 2, then their ratio would be 1 (ideal? As it shows orthogonal validation).

We thank the reviewer for this suggestion. We have now completely restructured the respective figure panels.

Perhaps for figure 4c and 4d, combining them into an UpSet plot (Venn diagram alternative) showing all shared and separate hits from each experiment, with a second UpSet plot showing the translation-annotated proteins? Just suggestions, since I don't find these hairballs to be very informative. (I do find the hairball on Fig 5b and Euler diagram on 5c to be more informative!)

We now use a combination of VENN diagram and listing of different groups to improve the clarity of this figure (Figure 4c, d).

Not a major concern, but just a note that this reviewer greatly appreciates the authors' deposition of data and code in the appropriate repositories! The submissions to ProteomeXchange appear complete, but I would request a file map (RAW file names to experimental metadata) be uploaded in addition to the excellent RAW and processed files that are currently deposited.

We updated our submission and added the requested file map.

REVIEWER COMMENTS

Reviewer #1 (Remarks to the Author):

In their revised version the authors have addressed all my previous concerns. They convincingly show that the C-terminal tail of Hsp104 is sufficient for nuclear targeting upon ageing and added all control experiments requested.

They also addressed the major concern and added microscopic analysis suggesting mobilization of nuclear-deposited translation factors (e.g. eIF2a) upon glucose addition in an Hsp104-dependent manner (Fig. 8d/e). This observation underlines the physiological relevance of the presented findings. However, the effect is only clearly seen in the quantifications (8e) but not so much in the provided images (8d), also because the authors added glucose to cells incubated for 48 h. The reviewer is wondering why the authors did not chose a 72 h sample as nuclear accumulation of Sui2-mCherry is much more pronounced (see Fig. 5f/g), which should make the effect more obvious and stronger. Similarly, enhanced aggregation of Sui2-mcherry in Hsp104* cells upon partial inhibition of the proteasome is not clearly apparent in the provided microscopic images (Fig. 8f). Here again pictures were taken after 48 h and the reviewer is wondering whether a longer incubation (72 h) linked to more pronounced nuclear accumulation of Sui2-mCherry will enhance the differences between Hsp104 wildtype and mutant cells.

Reviewer #2 (Remarks to the Author):

The authors have made substantial changes to the manuscript. They performed some key experiments that support their findings. The new data and explanations greatly improve the manuscript. Their changes are appreciated.

There remains one panel that wasn't addressed: Fig 7d (old) 7e (new). There is a question about the fact that luciferase folding isn't different at early timepoints with the Hsp104 mutant that can't enter the nucleus in aged cells. The authors conclusions about the effects of this mutant may not be well-justified if differences are only observed late after heat shock.

Reviewer #3 (Remarks to the Author):

The authors have sufficiently addressed my prior concerns regarding the data and conclusions made from quantitative proteomics techniques.

Point-by-point reply

Reviewer #1 (Remarks to the Author):

In their revised version the authors have addressed all my previous concerns. They convincingly show that the C-terminal tail of Hsp104 is sufficient for nuclear targeting upon ageing and added all control experiments requested.

They also addressed the major concern and added microscopic analysis suggesting mobilization of nuclear-deposited translation factors (e.g. eIF2a) upon glucose addition in an Hsp104-dependent manner (Fig. 8d/e). This observation underlines the physiological relevance of the presented findings. However, the effect is only clearly seen in the quantifications (8e) but not so much in the provided images (8d), also because the authors added glucose to cells incubated for 48 h. The reviewer is wondering why the authors did not chose a 72 h sample as nuclear accumulation of Sui2-mCherry is much more pronounced (see Fig. 5f/g), which should make the effect more obvious and stronger.

Similarly, enhanced aggregation of Sui2-mcherry in Hsp104* cells upon partial inhibition of the proteasome is not clearly apparent in the provided microscopic images (Fig. 8f). Here again pictures were taken after 48 h and the reviewer is wondering whether a longer incubation (72 h) linked to more pronounced nuclear accumulation of Sui2-mCherry will enhance the differences between Hsp104 wildtype and mutant cells.

We thank the reviewer for the constructive feedback. We performed these experiments both at 48 h and 72 h with comparable results, but have initially selected the 48-hour time point for consistency, as the majority of the mass spectrometry assays and microscopic analyses were conducted at 48 h. Following the reviewer's suggestion, we have now also included the quantification of nuclear accumulation of Sui2 at 72 h (Fig. 8e) and the corresponding representative widefield micrographs (Supplementary Fig.7f). Similarly, data for Sui2 aggregation at 72 h have been included (Supplementary Fig. 7g, h), albeit this has been performed with a reduced number of individual transformants. Across both experimental setups, we consistently observe comparable results for both time points.

In addition, we have reprocessed the widefield images using the "Unsharp-Mask" Tool in Fiji, as detailed in the Material and Methods section. For the micrographs in Figure 8d (and the newly included microscopic analysis at 72 h in Supplemental Fig. 7f), we now also present larger fields of view. For Figure 8f, we retained the narrower fields of view for better visibility of aggregates but reprocessed as described.

Reviewer #2 (Remarks to the Author):

The authors have made substantial changes to the manuscript. They performed some key experiments that support their findings. The new data and explanations greatly improve the manuscript. Their changes are appreciated.

There remains one panel that wasn't addressed: Fig 7d (old) 7e (new). There is a question about the fact that luciferase folding isn't different at early timepoints with the Hsp104 mutant that can't enter the nucleus in aged cells. The authors conclusions about the effects of this mutant may not be well-justified if differences are only observed late after heat shock.

We apologize, we might have misunderstood the reviewer's initial comment. We have now rephrased this paragraph to better describe the results. The assay measures total reactivation of denatured nuclear firefly luciferase, which includes species from near native conformations to deeply entangled/aggregated species. The Hsp70-system will reactivate many of these species while some are harder to reactivate and are in need of Hsp104 activity. Our understanding is that we have substrate selection going on during the recovery phase, so that near native conformations are reactivated rapidly (mainly by the Hsp70-system) and that nuclear Hsp104 disaggregation activity becomes more important later on when only the severely aggregated species remain due to the slow reactivation kinetics when disaggregation is involved. Since Hsp104* is not targeted to the nucleus, the disaggregation defect becomes apparent only in the later time points of the assay.

The revised paragraph on page reads as follows:

“Immediate reactivation of ^{GFP}FFL-NLS within 30 min after acute heat shock was comparable for Hsp104 and Hsp104 expressing cells. However, specifically in aged cells, wild type Hsp104 reactivated more heat-denatured ^{GFP}FFL-NLS than the nuclear targeting-defective Hsp104* within the recovery phase (Fig. 7d, e). Thus, initial reactivation of denatured ^{GFP}FFL-NLS was not impaired by the lack of nuclear Hsp104*, likely due to the dominant reactivation pathways executed by nuclear Hsp70 together with J-domain proteins/Hsp40s (Apj1) and Hsp110s (Sse1/Sse2)^{65,66,67}. Later on, when only the more severely aggregated species of ^{GFP}FFL-NLS remained, the lower levels of nuclear disaggregation activity in Hsp104* expressing cells became apparent. In line, hsp104Δ cells displayed impaired reactivation of ^{GFP}FFL-NLS immediately after heat shock, likely due a cell-wide loss of disaggregation activity and thus saturation of the Hsp70-40-110 axis.”*

Reviewer #3 (Remarks to the Author):

The authors have sufficiently addressed my prior concerns regarding the data and conclusions made from quantitative proteomics techniques.

REVIEWERS' COMMENTS

Reviewer #1 (Remarks to the Author):

The authors have sufficiently addressed my remaining concerns by adding additional timepoints of microscopic analysis. I therefore support publication of the study.

Reviewer #2 (Remarks to the Author):

The authors addressed my questions. Thank you.

REVIEWERS' COMMENTS

Reviewer #1 (Remarks to the Author):

The authors have sufficiently addressed my remaining concerns by adding additional timepoints of microscopic analysis. I therefore support publication of the study.

Reviewer #2 (Remarks to the Author):

The authors addressed my questions. Thank you.

We thank the reviewers for their constructive feedback during the revision and are happy to see that both reviewers support publication of our revised manuscript.